# Approaches in Animal Proteins and Natural Polysaccharides Application for Food Packaging: Edible Film Production and Quality Estimation

**DOI:** 10.3390/polym13101592

**Published:** 2021-05-15

**Authors:** Andrey Lisitsyn, Anastasia Semenova, Viktoria Nasonova, Ekaterina Polishchuk, Natalia Revutskaya, Ivan Kozyrev, Elena Kotenkova

**Affiliations:** 1Department of Scientific, Applied and Technological Developments, V. M. Gorbatov Federal Research Center for Food Systems of RAS, Talalikhina st., 26, 109316 Moscow, Russia; a.b.lisitsin@yandex.ru (A.L.); a.semenova@fncps.ru (A.S.); v.nasonova@fncps.ru (V.N.); n.revuckaya@fncps.ru (N.R.); iv.kozirev@fncps.ru (I.K.); 2Experimental Clinic and Research Laboratory for Bioactive Substances of Animal Origin, V. M. Gorbatov Federal Research Center for Food Systems of RAS, Talalikhina st., 26, 109316 Moscow, Russia; kat.1997@mail.ru

**Keywords:** biopolymers, edible films and coatings, dry and wet processes, mechanical properties, permeability, microstructure crosslinking

## Abstract

Natural biopolymers are an interesting resource for edible films production, as they are environmentally friendly packaging materials. The possibilities of the application of main animal proteins and natural polysaccharides are considered in the review, including the sources, structure, and limitations of usage. The main ways for overcoming the limitations caused by the physico-chemical properties of biopolymers are also discussed, including composites approaches, plasticizers, and the addition of crosslinking agents. Approaches for the production of biopolymer-based films and coatings are classified according to wet and dried processes and considered depending on biopolymer types. The methods for mechanical, physico-chemical, hydration, and uniformity estimation of edible films are reviewed.

## 1. Introduction

Inferior packaging or its absence causes significant food loss (about 20–25%) due to microbiological contamination and oxidative processes, which lead to a decrease in the quality of food products and makes them unsuitable for consumption [1].

The development and application of bioactive packaging systems is a relevant field of research. The application of such smart packaging systems is a tool for protecting food from spoilage and reducing the risk of growth of pathogenic microorganisms due to both the creation of a barrier and the action of active components at the border of the product with the packaging [2,3]. The currently used packaging materials are mainly produced from petrochemical products [4,5]. The global problem of environmental pollution makes alternative environmentally friendly and biodegradable polymers be in demand [6].

Every year, the problem of recycling polymer packaging materials becomes more acute [7] due to its accumulation in large quantities, which cause significant harm to the environment [8]. The incineration or pyrolysis of polymer waste, to some extent, solves the problem of their accumulation in landfills, but does not contribute to improving the overall environmental situation [9]. Recycling of polymer waste is more environmentally friendly, but in this case, significant labor and energy costs are required for sorting polymer materials and their subsequent processing [10,11]. It should be noted that the recycling of polymers is carried out a limited number of times, after which the problem of burial or incineration of these materials arises again [12].

Concerning environmental suitability, biopolymers are environmentally friendly packaging materials [13,14]. The main advantage of their use as bioplastics is the closed natural cycle, where the end of one cycle leads to the beginning of the next cycle [6].

Biopolymers can be divided into three main categories depending on the origin and method of production (Figure 1): directly extracted from biomass, synthesized bio-derived monomers, and produced by microorganisms [15]. Polysaccharides and proteins are the most promising biopolymers for the production of packaging materials [16,17]. Proteins are heteropolymers consisting of α-amino acids as monomeric units. The combinations of 20 amino acids to form a protein sequence allow for an almost unlimited number of various polymer chains with different physical and chemical properties. Proteins also contain a large number of functional groups that can be changed enzymatically, chemically, or physically for varying the properties of the films [16]. Polysaccharides are good candidates for replacing oil-based polymers due to their ability to form a film, affinity for paper-based materials, an appropriate barrier to gases and aromas, and good mechanical strength. Moreover, these biopolymers are biodegradable, non-toxic, and are used as a matrix for the inclusion of additives with specific functionality, such as active antimicrobial properties, for example [17].

The prefix “bio” indicates that biopolymers are biodegradable. The word “biodegradable” means that materials can be decomposed by bacteria, fungi, and yeast to the final products of biomass under anaerobic conditions—hydrocarbons and methane [20]. These types of polymers consist of monomers that are covalently connected, forming a chain of the molecule. They are also produced inside the cell as a result of complex metabolic processes. Biopolymers can be used for food packaging as a replacement for oil-based plastics made from petroleum due to their biodegradability, renewability, and wide distribution [21].

Environmental friendliness is a key ideology nowadays. The use of biopolymers from renewable resources could solve global plastic pollution. For many years, researchers have been trying to develop and design packaging materials based on natural biopolymers. However, animal proteins and natural polysaccharides are characterized by some undesirable properties caused by their chemical nature and structure. These disadvantages reduce their competitiveness with oil-based plastics but can be overcome. In this review, we summarize the description of such popular biopolymers as starch, cellulose, pectin, chitosan, alginate, casein, collagen and gelatin, considered for films and coatings production in general and for each biopolymer. The main characteristics of packaging materials based on animal proteins and natural polysaccharides, appropriateness for certain type of product based on properties of polymer, as well as ways to improve it are also described.

## 2. Biopolymers Used for Food Packaging

Biopolymers are widely implemented and can be used as coatings and films [4,22]. Coating involves the formation of a cover directly on the surface of food products, whereas films are structures that are used separately after formation [23].

Materials based on biopolymers must meet the basic requirements of health safety, mechanical and chemical resistance, and durability [24,25]. Therefore, food packaging should not only be biodegradable, but also functional. Compared to synthetic polymers derived from petroleum, biopolymers have a more complex chemical structure and side chain structure, which provides additional opportunities for the formation of packaging materials with specific characteristics for specific purposes [18,20].

Biopolymers directly extracted from biomass as polysaccharides and animal proteins, which are most often used for the preparation of packaging materials for food products, will be considered in this review [16,26,27].

### 2.1. Starch

Starch is one of the most readily available polysaccharides on the planet [28]. The plants from which it is obtained grow in almost all temperate climate zones. Corn, wheat, potatoes, and rice are the world leaders: 84%, 7%, 4%, and 1%, respectively [29]. This biopolymer is a mixture of amylose and amylopectin (Figure 2), the ratio of which varies depending on the type of starch.

The ratio of amylose:amylopectin varies significantly not only between different plants, but also within a single plant species or plant organ [31]. The conditions and phase of growth also influence this ratio [32,33]. Several studies have shown that a change in this ratio implies a change in the physical and chemical characteristics of starch and its interaction with other molecules, which leads to different swelling capacity [34], solubility in water, microscopic properties [35], and stability and barrier/mechanical properties in starch films [36]. According to some studies, starch with a high content of amylose, for example, from peas, has the best mechanical and gas barrier properties [37]. It was also noted that lentil starch (30% amylose) has a strong tendency to gelatinize at a relatively low concentration compared to corn and potato starch [29].

Cereal (grain) starch is obtained by its physical separation from non-starch components. Various processes of wet grinding of grains have been developed for the production of starch. The main stages of these processes are soaking, grinding, and separation of grain components [38]. Potato starch is extracted from potato tubers in a process called bio-processing: the potatoes are ground, and the contents of the cells are released, including starch and protein [39].

Starch-based polymers have low moisture resistance, which limits their use in packaging. The usual form of natural starch is a crystalline molecular structure that is not flexible [40]. However, an interesting starch derivative is thermoplastic starch (TPS), which is more convenient for films production [41,42], which could be obtained with thermal and physical impacts in the presence of plasticizers [40]. Various physical and chemical reactions are involved in the heat treatment of starch-based polymers, such as water diffusion, granule expansion, gelatinization, melting, crystallization, and extrusion [43].

TPS with improving properties can be used in the field of food packaging since it is economical and available in large quantities. The production of flexible and solid packaging (biofilms, bags, laminated plastic, etc.) is the main sector of TPS application in the food industry [40]. Polymer films derived from starch are biodegradable and have good properties as oxygen barriers. However, the amount of plasticizer, humidity, and amylose content are the limiting factors that determine the mechanical properties of TPS. The type and amount of plasticizer used in the production of thermoplastic starch strongly affect the physical, chemical, and thermal properties of the film [44]. Various structural enhancers, such as microcrystalline cellulose, carboxymethylcellulose, carbon nanotubes, etc., are added to the starch-polymer matrix in order to improve its properties [45]. Various types of such reinforced starch are already successfully used in the packaging of bread, vegetables, and meat products stored in standard conditions [41,46,47].

### 2.2. Cellulose

Cellulose is the most common natural biopolymer and consists of β-(1–4)-D-glucopyranose monomers (Figure 3) [48]. It is biosynthesized by a number of living organisms, from lower to higher plants, marine animals, bacteria, and fungi [49]. It has been estimated that 1011–1012 tons are synthesized annually by photosynthesis in a fairly pure form, for example, in the seed hairs of the cotton plant, but mainly cellulose combines with lignin and other polysaccharides (hemicelluloses) in the cell wall of woody plants [50]. Although cellulose is primarily found in forests, where wood is the most important source, cellulose-containing materials include agricultural residues, aquatic plants, grasses, and other plant matter [51,52]. Commercial cellulose production is based on harvested sources, such as wood, or on natural sources with high biopolymer content, such as cotton [53]. In contrast with starch, cellulose is a linear polymer without winding and branching [48]. Numerous hydroxyl groups in cellulose form strong hydrogen bonds, which make the material non-fusible. Chemical modification of cellulose is required for the production of flexible materials, which often involves the replacement of hydroxyl groups with acetate or methyl groups (esterification), the purpose of which is to reduce the intensity of the hydrogen bonds [54]. The rate of esterification and type of replacers, as well as the length of the polymer chain, affect the further permeability, mechanical properties, and solubility. Methylcellulose (MC) and carboxymethylcellulose (CMC) are the most common cellulose esters and have good film-forming properties, which allow them to be used as packaging materials for food products [55,56,57,58].

Methylcellulose (MC) is formed when one or more hydroxyl groups (-OH) in the anhydroglucose are replaced by a methoxide group (-OCH_3_). The degree of substitution of MC ranges from 1.4 to 2.0, and it is soluble in cold water [59]. It is reported that MC forms unbreakable, flexible, transparent, tasteless, and non-toxic films that have good barrier properties for oxygen, but bad ones for water vapor [60]. MC has a relatively high tensile strength and elastic modulus, so it demonstrates a reinforcing effect [61] and exhibits improved mechanical properties in mixtures containing proteins, lipids, etc. [62].

Carboxymethylcellulose (CMC) is a cellulose derivative in which some hydroxyl groups of glucopyranose units in cellulose are replaced by carboxymethyl groups. CMC is formed by reaction with chloroacetic acid (ClCH_2_CO_2_H) and catalyzed by alkali [63]. The amount of carboxymethyl groups in the cellulose molecule improves the strength of CMC film due to strong intermolecular forces [64]. Initially, CMC was studied as a hydrogel polymer, but it was later discovered that its dry form could be considered a biodegradable alternative to petroleum-based food packaging materials [64]. CMC easily absorbs moisture, dissolves in cold water, and exhibits thermal gelatinization [65]. CMC-based films can be combined with MC, clay, chitosan, etc., which usually improves the mechanical properties of the film: the tensile strength and elastic modulus increase, while the strain at break of the films is reduced [65]. It was noted that such a plasticizer as glycerin significantly increases the flexibility of the film, but also reduces the tensile strength and elastic modulus [66].

Cellulose derivatives are more suitable for packaging that is in direct contact with food products [65]. Studies of the properties of various cellulose derivatives used for film formation are continuing with the aim of developing mixtures of cellulose derivatives with other biopolymers to improve the mechanical, barrier properties and increase the shelf life of packaged food products [46,67,68].

### 2.3. Pectin

Pectin is one of the main components of the plant cell wall, contributing to the integrity and rigidity of tissues, and is considered one of the most complex macromolecules in nature [69]. Although pectin is ubiquitous in the plant kingdom, pectin derived from apples, citrus fruits, sunflowers, and sugar beets is an undisputed commercial source for the processing industry due to their physical and chemical properties and the availability of biomass [70]. Pectin is a poly–α–1–4–galacturonic acid (Figure 4a) with different degrees of methylation of carboxylic acid residues and/or amidated polygalacturonic acids (Figure 4b) [71]. Carboxyl groups of galacturonic acid are esterificated with methanol, resulting in methoxylated carboxyl groups. On the other hand, amidated carboxyl groups are obtained when galacturonic acid is converted with ammonia to carboxylic acid amide.

According to the degree of esterification by methanol (the ratio of esterified galacturonic acid groups to its total number), pectin can be classified as high-methoxylpectin (HMP, >50% of esterified carboxyl groups) pectin or low-methoxyl pectin (LMP, <50% of esterified carboxyl groups) [70]. The degree of esterification affects the gelling properties of pectins. For example, pectin with a low content of methoxyl groups forms a gel in the presence of multivalent ions, which bonds pairs of carboxyl groups of different pectin chains. Pectin with a high content of methoxyl groups forms a gel in acidic solutions with the addition of various sugars, such as sucrose or glucose [72,73]. LMP is used for encapsulation, food packaging film processing, and low-calorie gels in dietary foods [72]. Structurally, pectins are classified as rhamnogalacturonan-I (RG-I; “hairy” pectin), substituted galacturonans (RG-II or SG; “hairy” pectin), and homogalacturonans (HG; smooth pectin). The hairy region of pectins (RG-I and RG-II) has a high probiotic potential [74]. Thus, based on the macromolecular and microstructural characteristics of pectins, the scope of their application in food products is determined.

Pectin is an ingredient used in the food industry without any restrictions other than current good manufacturing practices, is generally recognized as safe (GRAS) by the U.S. Food and Drug Administration (FDA), and is used in food primarily as a gelling agent, stabilizing agent, or thickener in products such as jams, yogurt drinks, fruit milk drinks, and ice cream [75,76]. The ubiquitous presence, low cost, structural flexibility, and polymerization ability of pectin contribute to its use as a matrix for active food packaging materials [70]. Since bioactive packaging films made from pectin have very weak antimicrobial properties, their antimicrobial potential can be enhanced by integrating and combining them with various functional compounds, such as essential oils, phenolic compounds, nanomaterials, free fatty acids, and others [77]. The production of edible films from pectin can be carried out in various ways, such as casting, extrusion, spraying, and coating with a knife [78].

### 2.4. Chitosan

Chitosan is a linear polysaccharide consisting of randomly linked units of β-(1,4)-D-glucosamine and N-acetyl-D-glucosamine (Figure 5). Chitin is the second most common structural polysaccharide found in nature after cellulose and is usually deacetylated by an alkali for chitosan production [79,80]. Natively, chitin is presented in the form of ordered crystalline microfibrils that form structural components in the exoskeleton of arthropods or in the cell walls of fungi and yeast. So far, the main commercial sources of chitin are the shells of crabs and shrimps [81]. In industrial processing, chitin is extracted by acid treatment to dissolve calcium carbonate, and then by an alkaline solution to dissolve proteins. In addition, a discoloration stage is often added to remove the pigments and obtain a colorless pure chitin [82]. Chitin is usually isolated from the exoskeleton of crustaceans and, in particular, from shrimps and crabs, where α-chitin is produced [83]. Squid is another important source of chitin, in which it exists in the β-form, which has been found to be more amenable to deacetylation. Such chitin also shows higher solubility, reactivity, and affinity for solvents and swelling than α-chitin due to the much weaker intermolecular hydrogen bond attributed to the parallel arrangement of the main chains [83]. Some types of insects, fungi, bacteria, and algae can also be alternative sources of chitin/chitosan [84,85,86].

The term “chitosan” usually refers to a family of polymers produced after the deacetylation of chitin to varying degrees [81]. In fact, the acetylation degree, which reflects the balance between the two types of residues, distinguishes chitin from chitosan. When the acetylation degree is higher 60%, the product is called chitosan and becomes soluble in acidic aqueous solutions [88]. A depolymerization reaction also occurs during deacetylation, as evidenced by changes in the molecular weight of chitosan. Chitin can be converted into chitosan by enzyme treatment [89] or by a chemical process [90]. Chemical methods are widely used for commercial purposes of producing chitosan due to their low cost and suitability for mass production [81]. Chitosan has been found to be non-toxic, biodegradable, bio-functional, and demonstrates good film-forming qualities and antimicrobial properties [91].

However, films prepared only from chitosan do not meet the criteria for packaging materials. They are rigid and brittle; therefore, it is important to use plasticizers to improve their mechanical properties. Various plasticizers have an effect on the stability of plasticized films. It was found that polyethylene glycol (PEG) and glycerin are the best candidates as plasticizers of chitosan films [92]. It was found that other small-molecular organic materials also improve the mechanical properties of chitosan films, but their effectiveness decreases within a few weeks [92]. The addition of PEG and glycerol promotes the formation of relatively flexible and stable (at least for several months) chitosan films. In general, these substances prevent the formation of double bonds between adjacent polymer chains, which prevents recrystallization [93]. Plasticizers modify the mechanical properties of chitosan without changing its fundamental chemical structure [92].

In comparison with other biopolymers used for food packaging, chitosan has an advantage due to its ability to include such functional substances as minerals or vitamins and its antibacterial activity, which is important for maintaining the quality of products [94,95,96].

### 2.5. Alginate

Alginates are natural indigestible polysaccharides, usually produced and purified from various genera of brown algae (mainly *Laminaria hyperborean*, *Macrocystis pyrifera*, *Ascophyllum nodosum*, and to a lesser extent, *Laminaria digitate*, *Laminaria japonica*, *Eclonia maxima*, *Lesonia negrescens*, *Sargassum sp.*) [97]. Some bacteria, such as *Azotobacter vinelandii* or mucoid strains of *Pseudomonas aeruginosa*, also synthesize alginate-like polymers as exopolysaccharides [98,99]. *Azotobacter vinelandii* is a strict aerobe that can fix nitrogen and synthesize two polymers during vegetative growth: alginate and intracellular polyesters (polyhydroxybutyrate) [100]. *A. vinelandii* produce alginate during the encystment process as a mechanism for increasing resistance to drying in adverse environmental conditions to maintain cell hydration, being its structural element [101]. Moreover, it can act as a protective barrier against heavy metal toxicity, creating an ion exchange system with selectivity for Ca^2+^ [102]. *P. aeruginosa* (mucoid phenotype) produce alginate as the causative agent of cystic fibrosis, and it is associated with pathogenicity. Alginate biosynthesis in *Pseudomonas spp*. is induced under dehydration conditions and is probably a key component of stable biofilms in various media [103]. 

The molecular structure of alginates consists of unbranched linear binary copolymers of β-D-mannuronic acid (M) and α-L-guluronic acid (G) residues bound by 1–4 glycosidic bonds (Figure 6) [97]. The structure of alginate algae can be divided into three fractions (three blocks of uronic acid): these are the homopolymer regions of the M and G blocks and the alternating MG blocks containing both polyuronic acids [104]. Bacterial alginates have O-acetyl groups, while in the structure of alginate algae, they are absent [105]. In addition, bacterial alginates have a higher molecular weight compared to algae polymers [106]. The source of alginate affects the ratio of M and G residues, which has an impact on the physical and chemical properties of alginate, as well as on solution viscosity and thickness of the film [97].

The solubility of various types of alginates in numerous solvents and solutions was indicated by Kimica Corporation [107]. The FDA classifies food grade sodium alginate as a GRAS (generally considered safe) in the Code of Federal Regulations (CFP) and list its use as an emulsifier, stabilizer, thickener, and gelling agent [108]. The European Commission (EC) has added alginic acid and its salts (E400–E404) to the list of permitted food additives [109].

Alginate is widely used in various industries, such as food, beverage, textile, printing, and pharmaceutical, as a thickener, stabilizer, emulsifier, chelating agent, encapsulator, suspending agent, or used for the formation of gels, films, and membranes [110]. About 30,000 metric tons of alginate derived from brown algae are produced annually [100]. Alginate is known for its biocompatible and biodegradable properties, as well as its low price. The ability of functional groups of G blocks to interact with polyvalent cations (for example, Ca^2+^, Al^3+^, and Fe^3+^) is an important characteristic of alginate [111]. Among divalent ions, calcium ions usually react with alginate to form a polymer with low solubility. In general, the length of the G-block characterizes the ability and selectivity of the alginate in the formation of these interactions, whereas the M and MG blocks have almost no selectivity. The M and G blocks bind via ions to form a three-dimensional structure («egg-box») [112]. This triggers the process of anionic exchange, in which the water-soluble alginate exchanges its counter ions for Ca^2+^. This ionic crosslinking leads to the formation of cold-setting and heat-resistant films. Alginate films are cross-linked to improve their water resistance, mechanical properties, and coherence [111].

### 2.6. Casein

Caseins are suitable hydrocolloids for the formation of edible films among proteins due to their high nutritional value, solubility in water, and ability to emulsify. Casein is the main protein extracted from milk (~80%) [113], consisting of four different protein fractions: α_S1_-, α_S2_-, β-, and κ-casein [114], which together form colloidal micelles in milk (Figure 7) stabilized by casein structures and calcium-phosphate bridges [115]. The unique properties of the four protein fractions affect the film-forming ability of casein [116].

The global production of caseins and caseinates is difficult to determine due to the lack of significant data. The largest casein producers are New Zealand (150,000 tons), Netherlands (85,000–10,000 tons) and Germany (25,000–40,000 tons). The global market for casein or caseinates used in the food industry ranges from 200,000 to 2,500,000 tons [118].

Casein is precipitated from skimmed milk by acidifying it to produce acidic casein, or the milk is treated with rennet to produce rennet casein. The precipitated casein clot is separated from the serum, washed, and dried [119]. Caseinates are water-soluble derivatives of acidic caseins formed by reaction with alkalis. Edible casein is a dairy byproduct that is used as an ingredient in many foods, including dairy products [120]. The development of food technologies and their applications has increased the production of casein and the demand for it. Its production differs from that of non-food casein (also called industrial casein) because food casein is produced under sanitary conditions [118]. In addition, food-grade chemicals are used for its production that undergoes sufficient heat treatment to ensure that casein is safe for human consumption [118]. Intensive research on production technologies over the years and their implementation into factories has significantly improved the approaches for food-grade casein production.

Depending on the coagulation method, different types of casein with certain characteristics are obtained; rennet and acid casein are the main available types [121]. Acidic casein is insoluble in water, has a pH of about 4.6, and is precipitated by acidifying milk with mineral acids or lactic acid [113]. Rennet casein is insoluble in water, has a pH of about 7.5, and is coagulated by the action of chymosin, which cleaves the k-casein tail and causes destabilization of the casein micelles [114]. Acidic casein can be solubilized by neutralization with alkali, such as sodium, potassium, or calcium hydroxide, to produce sodium, potassium, or calcium caseinates, respectively. This caseinate does not contain colloidal calcium phosphates, has a pH close to 7, and is highly soluble in water [122].

Casein films have good barrier properties for oxygen and other non-polar molecules due to the distribution of polar amino acids along the protein chain, which allows them to protect products that are prone to oxidation [122]. On the other hand, the interaction forces between the polar and non-polar amino acids in the casein structure form a cohesive film matrix that tends to shrink during drying and becomes brittle [123]. Food-grade plasticizers, such as glycerin or sorbitol, are added to the film-forming solution in order to solve this problem. Plasticizers increase the thermoplasticity of the protein film but reduce its strength [124].

Although casein films have potential for use as food packaging, some disadvantages need to be addressed before they can be widely used for commercial purposes. Casein-based films are very sensitive to moisture, they absorb and release water molecules, which greatly affect their mechanical and barrier properties, and moreover, they are mostly soluble in water, which limits their areas of application [125]. It is also worth noting that plasticized casein films cannot provide high mechanical strength or good elasticity compared with synthetic polymer materials. By crosslinking polymer chains, chemical and physical processing can be used for modification of the polymer matrix, which improves the functionality of the protein film [126]. Glutaraldehyde or the natural compound genipin, as well as transglutaminase, etc. are typical chemicals used as crosslinking agents [114,127,128]. In general, aldehydes are quite effective as binding agents of protein molecules, but their high toxicity is unacceptable for foods. Genipin and transglutaminase are known to be safe, but their high cost limits their further use in industry [114]. Thus, the search for new cheap crosslinking agents safe for use in the food industry is underway.

### 2.7. Collagen

Collagen is the most abundant protein in the extracellular matrix of vertebrates, accounting for approximately 30% of the total body protein mass. It is absent in plants and unicellular organism, where its role is played by polysaccharides and cellulose. Collagen is presented in the body walls and cuticle in invertebrates, while in mammals, it is found in the cornea, bones, blood vessels, cartilage, dentin of teeth, etc. [129].

Bovine hides are a by-product of meat production and mainly used for the production of leather, but the inner corium layer of the hide is rich in collagen [130]. This collagen has a higher denaturation temperature compared to collagen from marine sources [131]. Collagen can also be extracted from fish and pigs, but there are some limitations: the use of fish collagen is limited because its low hydroxyproline content [132] gives collagen a low denaturation temperature, while pork collagen is restricted due to religious considerations [133].

Collagen consists of three parallel chains of α-helices twisted in the form of a right triple helix [129] and constructed from frequently repeated fragments with a characteristic sequence -Gly-X-Y- (Figure 8). Glycine is every third amino acid residue, proline is often found in the X position, and the Y position can be occupied by both proline and 4-hydroxyproline. In addition, the collagen molecule contains residues of 3-hydroxyproline and 5-hydroxylysine. The right triple helix self-associates to form highly ordered cross-linked fibrils [129]. These fibrils form insoluble fibers that provide an extracellular matrix with high integrity and mechanical tensile strength due to its tightly wound triple helical structure. Collagen molecules are divided into 26 different types, which are grouped into 8 families, depending on its structure, chain bonding, and position in the organism. These families include fibril-forming, basement membrane, anchoring fibrils, microfibrillar, hexagonal network-forming, transmembrane, multiplexins and fibril-associated collagens with interrupted triple helix [134].

Water-soluble collagen presents in a small percentage of total collagen. The solubility of collagen depends on the type of tissue and age. A neutral salt solution or dilute acetic acid are the most commonly used solvents for collagen extraction. Strong alkali or enzymes are used to extract the insoluble collagen to break down the additional cross-linked bonds [136]. Collagen can be used for the production of edible films in the meat industry, e.g., for sausages, salami, snacks [136], due to its good mechanical properties [137]. Fibrillar collagens can easily form stable films capable of contracting and stretching to adapt to the compression and expansion of meat raw materials during continuous processing [138,139]. Collagen films can become an embedded/edible part of meat products so they can provide safety benefits, control quality changes, and reduce the loss of shrinkage of meat and meat-based products during storage, thereby extending the shelf life and maintaining the visual appeal of products for a long time [140].

### 2.8. Gelatin

Gelatin is a naturally occurring water-soluble protein characterized by the absence of a noticeable odor and the random configuration of polypeptide chains in an aqueous solution. It is obtained by partial hydrolysis of collagen [141]; its structure is rigid rod-shaped molecules that are located in fibers connected by covalent bonds [142].

Pig skin was used as a raw material for the production of gelatin in the 1930s and continues to be the most important material for the large-scale industrial production of food gelatin [143]. For more expensive applications, such as pharmaceuticals, gelatin is usually obtained from cattle bones, which is considered a more complex and expensive extraction process [144]. However, in an effort to replace pork and bovine gelatin, fish gelatin production has increased over the past decade, accounting for more than 1.5% of total gelatin production. In recent years, by-products obtained in the fishing industry, such as heads, skin, bones, fins, muscle parts, scales, internal organs, and others, are considered as potential sources of gelatin and not recyclable waste [145]. The main disadvantages of fish gelatin are its rheological properties, since it is less stable than mammalian gelatin. Moreover, since the production of gelatin from fish and poultry is still limited, the resulting products are less competitive in price than products made from mammalian gelatin [146,147].

Gelatin is a heterogeneous polypeptide mixture of α-chains, β-chains, and γ-chains (Figure 9).

Gelatin can be divided into two types depending on the processing method. Type A has an isoelectric point at pH ~ 8–9 and is obtained from acid-treated collagen. Type B has an isoelectric point at pH ~ 4–5 and is obtained from an alkali-treated collagen, during which asparagine and glutamine residues are converted to their respective acids, resulting in a higher viscosity [149]. Gelatin obtained from pig skin is commonly referred to as type A, and gelatin obtained from beef skin or cattle skins and bones is referred to as type B.

Gelatin has various functional properties that can be divided into two groups: properties related to the external surface, such as protective colloidal function, formation and stabilization of emulsion and foam, adhesion and cohesion, as well as the ability to form films and properties related to gelation, such as thickening, texturing, and water-binding ability [150,151,152]. Thus, a wide range of gelatin applications can be used in the food, packaging, pharmaceutical, cosmetic, and photographic industries, but the limited thermal stability and mechanical properties of gelatin, especially during processing, limit its implementation [139].

The film-forming properties of gelatin are widely used as an outer film to protect food products from drying out, exposure to light and/or oxygen during their shelf life [153]. Due to the high hygroscopicity of gelatin, it tends to swell or dissolve upon contact with the surface of foods with high moisture content [154].

Several scientific studies have been conducted to evaluate the overall effect of adding crosslinking and strengthening agents, plasticizers, or additives with antimicrobial or antioxidant properties to gelatin-based films to improve its functional properties and increase the shelf life of food products [155,156]. The improvement of these properties occurs when the intermolecular interactions of the protein chains decrease under the influence of molecular structures that modify their hydrophilic character or promote the formation of strong covalent bonds in the protein film [157].

Zhao et al. demonstrated the feasibility of using natural extracts as new natural crosslinking agents for the modification of gelatin (type B, from bovine bone) by forming hydrogen bonds between water and free hydroxyl groups of amino or polyphenolic groups, which significantly increased the strength of the gel compared to untreated gelatin [158]. Combining gelatin with other biopolymers, such as whey proteins, starch, chitosan, or pectin, can also be a good way to improve the mechanical properties and water resistance of gelatin-based films [159,160,161,162].

## 3. Approaches for the Production of Biopolymer-Based Films and Coatings

The film production is a physico-chemical process based on the intermolecular interaction of the biopolymer with the components of the solution in order to create a stable polymer matrix [163]. The process of film formation depends on the ability of the biopolymer to form a continuous crystalline or amorphous structure; therefore, at the initial stage, it is important to determine the type of polymer, taking into account its properties and the features of the mechanism of structure formation [164].

The film formation from polysaccharides is based on the breaking of polymer chains and the creation of new hydrophilic, hydrogen bonds and Van der Waals forces due to electrolytic and ion crosslinking [165]. Protein-based biopolymers, compared to other film-forming materials, have distinctive features: conformational denaturation of the protein (changes in secondary, tertiary, and quaternary structures), the presence of electrostatic charges, and amphiphilic nature (the presence of hydrophilic and hydrophobic molecules) [166]. The mechanism of formation of films from protein consists of its denaturation under the influence of acid/alkaline, mechanical, or enzymatic treatments, heat treatment, and various solvents, which leads to the formation of new molecular bonds between the peptide chains—ionic and hydrogen bonds, Van der Waals forces, hydrophobic interactions, covalent polar and nonpolar bonds-disulfide bridges, and cross-linking [116,167,168,169]. The molecular interaction of the protein-polysaccharide complex occurs on the basis of Van der Waals and electrostatic forces, hydrophobic and hydrogen bonds, the effects of excluded volume (the balance between steric repulsion of chain links and effective short-term attraction), or through chemical binding, for example, the Maillard reaction—the interaction of amino acids with sugars, including fructose, glucose, etc. [170].

The films and coatings are obtained by dissolving, dispersing, and emulsifying the biopolymer in a specific solvent. According to the safety and biodegradability requirements for food films and coatings, water, ethanol and their mixtures, and organic acids are used as solvents [166]. Magnetic or electric stirrers are used for acceleration of the dissolution of biopolymers [171,172], and a vacuum pump or ultrasound equipment are used to remove air bubbles from the solution [173].

### 3.1. Methods of Forming Edible Films 

Edible food films are usually made in two ways: wet and dry processes, using the appropriate methods (Figure 10) [174].

In the wet process, the film-forming components are mixed with a solvent, followed by drying the solution to obtain a food film [175]. The dry process consists of the formation of a film by heat treatment of a film-forming composition with a low solvent content [176]. In both processes, the film composition may include a single biopolymer or a combination of different biopolymers [177]. The wet process is mainly represented by the casting method [178], while the dry process is more variable—extrusion, injection molding, compression molding, co-extrusion, blown-film extrusion, and others [179,180,181].

Also, the films can have a mono-, di-, or multi- layer structure [182]. The production of di- and multi- layer films is carried out using the “layer by layer” technology, alternating solutions of biopolymers [183]. This method of film production is less common in the food industry because it is more labor-intensive, despite the fact that the finished films have improved barrier and mechanical characteristics [184].

#### 3.1.1. Casting Method

The most common method of forming edible films is the casting method (Figure 11) referring to the wet process [185]. This method is widely used in laboratory and pilot scales because it is quite simple and does not require special equipment. The film production process includes 3 main stages: (1) solubilization of the biopolymer in the solvent; (2) casting of the solution in the mold; (3) drying of the solution [186,187].

Various materials with low adhesion are used as forms for drying film-forming solutions. Teflon [188], glass coating (can be coated with a polyester film) [189], Petri dishes [190], Plexiglass [191], and polycarbonate plates [192] are the most common. Films are dried in various ways: under ambient conditions, with hot air, infrared energy, or microwave energy, which influence the formation of physico-chemical and mechanical properties [193].

The continuous casting method allows the films to be formed through the continuous flow of the biopolymer solution to a conveyor belt and then dried. The advantage of this method is the ability to choose such modes as the speed of the conveyor and the casting cycles of the solution, thereby adjusting the film thickness, temperature, and duration of drying [194]. This method makes it possible to obtain multilayer films [182].

The casting method is one of the most common. Dai et al. [195] studied the properties of films based on native and modified starches from various plant sources. Bagheri et al. [196] examined the effect of drying temperature on films prepared from sodium alginate with glycerin. Films plasticized with glycerin were cast using chitosan of two different molecular weights [197] and cellulose sulfate [198]. Liu et al. [199] investigated casting parameters for a mixture of gelatin, glycerin, and transglutaminase.

#### 3.1.2. Extrusion Method

Extrusion is a continuous technological process of obtaining a film material by pushing the molten film-forming composition through a forming hole of a certain shape (Figure 12) [200]. This process involves several operations simultaneously: melting, mixing, dosing, and forming [201].

The method is based on the thermoplastic properties of polymers during plasticization and heating above their glass transition temperature under conditions of low water content [202]. Therefore, the thermoplastic properties of film-forming materials must be well studied for the production of films in this way. It is necessary to take into account the influence of plasticizers that lower the glass transition temperature and any other additives on the thermoplasticity of film-forming materials [174].

The production of films by extrusion includes 3 main stages: (1) loading the film-forming composition in the form of a gel, powder, or concentrated solution into the feed hopper; (2) adding the composite components and mixing; (3) heating and applying to a conveyor belt or drum, followed by cooling on metal shafts [203]. The structure of biopolymer films becomes more homogeneous during the extrusion process, as well as their physicochemical, mechanical, and barrier properties are improved [204,205]. 

Processing parameters such as die temperature, die construction, speed, and number of screws directly affect the properties of the film [206]. The construction of the dies is chosen depending on the viscosity of the solution and the required film thickness: (1) sheet dies or slotted; (2) flat-film designed for blown-film dies most often annular, less viscous solutions are suitable for them; (3) pipe and tubing dies (annular); (4) profile extrusion dies (all possible geometric shapes—square, U and T-shaped); (5) dies for co-extrusion dies [207]. Single- or twin-screw extruders are used in the laboratory and industry to mix the ingredients evenly [208]. The optimal properties of the film also depend on the speed of the screw.

González-Seligra et al. [209] noted that, at a screw speed of 40 rpm, the starch/glycerol films were characterized by a large amount of non-gelated starch and an uneven distribution of glycerol. At a speed of 80 rpm, more homogeneous films were obtained, but with rapid retrogradation of starch (the transition of starch from a soluble state to an insoluble state, leading to an increase in the rigidity and brittleness of the film) [210]. At a speed of 120 rpm, thermoplastic films with good mechanical properties (tensile strength) and slower starch retrogradation were obtained. Calderón-Castro et al. [211] used pretreatment with a double-screw extruder with a speed of 120 rpm and a temperature of 100 °C with a cylindrical die and 3 heating zones to cast the extruded composition of physically modified starch. Compositions of pectin and starch with glycerol added in various combinations were extruded using a twin-screw extruder with a slit matrix to produce a thin film, which was dried in vacuum at room temperature before testing [212]. A twin-screw extruder with a cylindrical die was used to produce films from pectin and gelatin/sodium alginate [213]. A dry method was developed for the production of film sheets from rennet, acid casein, and sodium caseinate using a twin-screw extruder at a speed of 170 rpm and a temperature of 10–75 °C in various zones, and glycerin was added to the second zone; films from rennet casein, waxes, glycerin, and potassium sorbate were obtained by the same principle [122,214]. Collagen gels with plasma proteins, soy protein isolate, and gluten were processed using a single-screw extruder at 65 rpm, with saturated sodium chloride solutions at the exit of the annular die to remove the remaining moisture [215].

Blown-film extrusion is an alternative approach to achieve the industrial production of films, often using a single-screw extruder with an annular die and pressure shafts to pull the blown film [216]. This method is also used for the production of films from various biopolymers and their combinations: thermoplastic starch with plasticized chitosan processed in a twin-screw extruder to produce pellets with further blowing on a single-screw mechanism with a ring head [217]; film blown from granules of plasticized sodium caseinate [218]; granules of starch and gelatin reinforced with cellulose, pre-made by compression molding at 100 °C using a single screw for subsequent blow molding [219].

Co-extrusion is the process of simultaneous extrusion of two or more materials to form a multilayer film or coating [220]. This method is also used for applying an edible film/coating to food products, for example, a film of calcium alginate formed on the surface of minced meat when they simultaneously pass through a co-extrusion die [221]; alginate films with various polysaccharides (carrageenan, pectin, starches, cellulose, and gums) applied for sausage products packaging [222]; sodium alginate films with additives used for packaging sausages made by dry fermentation [223]; collagen films [224].

#### 3.1.3. Compression Molding

Compression molding is one of the oldest and most convenient methods of material processing [225]. The main stage of this process consists of heating the material under high pressure in the mold until it solidifies as a result of the formation of new bonds of biopolymer chains [226]. 

The extrusion method is often used to prepare the material before the main thermoforming process. Krishna et al. [227] obtained films from fish gelatin with glycerin by double-screw extrusion with further compression molding on teflon plates. Ceballos et al. [228] treated manioc starch with glycerin obtained on a double-screw extruder with a temperature-controlled hydraulic press. The films could be produced in one stage. A mixture of fish gelatin solution with glycerin and citric acid was subjected to compression molding using the caver laboratory press [229], as well as the composition of corn starch with glycerin and citric acid, previously mixed in a two-roll mill, followed by pressing [230]. Between the press and the film, sheets of various materials are used, for example, polytetrafluoroethylene [228], steel [230], aluminum [225], etc.

#### 3.1.4. Injection Molding

Injection molding is a widely used process for mass production of plastic products [231], but it can also be used for the production of edible films and consists of three main stages—filling, packing, and cooling [232]. Pre-injection pressure temperature, injection pressure, and molding temperature are the most important parameters [233].

Yu et al. [234] applied the method of twin-screw extrusion for the subsequent formation of granules from plasticized wheat starch with or without polyethylene oxide. At the second stage, injection molding was performed to produce films. Cho et al. [235] demonstrated the ability to process wheat gluten and glycerin pellets on a three-phase screw injection molding machine by the compression molding method. Pea isolate films with glycerol addition were made using the Injection Molding System [233].

### 3.2. Methods of Forming Coatings

Edible coatings of food products could be prepared in different approaches: dipping, spraying, brushing, fluidized bed, and panning methods (Figure 13) [236].

The use of a certain method depends on the type of coated product [237,238]. The feasibility of using coatings is explained by the following functional features: barrier properties to gases (O_2_, CO_2_, C_2_H_4_) and water, mechanical and optical properties, homogeneity, appearance, thickness, uniform distribution of the solution (wettability) on the surface, and adhesion (interaction between the surface of the product and the coating) [239]. 

The fluidized-bed and panning methods are difficult to perform and constitute in ways of spraying the film-forming solution: in the first case, it is produced on the surface of the fluidized bed and in the second case, it is sprayed on the processed product placed into a rotating pan [240,241]. Brushing is a method of applying the biopolymer solution with a brush [239]. Dipping and spraying are the most common methods due to simplicity and the low cost of the equipment [178]. The dipping method is most commonly used in the laboratory and is well suited for products with good surface adhesion [242], while spraying is convenient for implementation in industry [202].

#### 3.2.1. Dipping Method

Dipping is a method of coating in which the product is immersed in a film-forming solution for a certain time, removed, and left to form a film on the surface of the product (Figure 14) [236]. If one dip is not enough, since part of the solution may not be completely distributed over the surface of the product, this procedure is repeated [243]. The properties of the coating depend on the density, viscosity, and surface tension of the solution [244].

The disadvantage of the method is the possibility of forming an uneven and thick coating layer, which affects the appearance of the product, its shelf life, mechanical characteristics, and gas permeability [244]. Fruits are covered with a fairly thin layer, while vegetables and meat products have a thicker layer [245]. There are also a number of problems, such as possible dilution of the solution with pieces of product, resulting in the accumulation of product residues and the development of microbes in the immersion bath. This method is more suitable for shapeless products [246].

The dipping method is used for coating melon slices with alginate solution [247], sliced pears with chitosan and/or carboxymethylchitosan solutions mixed with sodium chloride [248], rainbow trout fillet with sodium alginate solution [249], fresh kiwifruit with sodium alginate solution mixed with calcium chloride as a cross-linking agent [250], and papaya with k-carrageenan solution plasticized with glycerin [251].

#### 3.2.2. Spraying Method

The spraying method (Figure 15) is carried out using spray equipment, from which the solution is applied to the surface of the product under pressure using special nozzles. The quality of the coating depends on a set of these parameters [178]. This method is suitable for fluid solutions, since more viscous ones make it difficult to spray [97]. The spraying method allows the product to be evenly coated, the thickness of the coating to be controlled, the temperature of the solution to be controlled, and large surfaces of products to be covered, but can have large losses of the solution to the ambient surroundings [177].

There are several main types of spraying. Air spray atomization is the method when the film-forming solution is sprayed under low pressure mixed with atmospheric air. Pressure atomization is carried out without air and under high pressure. Air assisted airless atomization is a combined type of atomization, which consists of feeding the solution under high pressure and flattening with air at the last stage of processing for a better distribution of the composition on the surface of the product [178,252]. The latter two methods can be used for more viscous solutions, but the equipment for them has a high cost [186].

Electrostatic spraying is a modern method, which is spread enough [253]. An electric charge is given to small particles of the film-forming solution, which is moved under the influence of an electric field [254]. The main difference from typical spraying methods is a uniform coating layer and an increase in the adhesion on the product surface, which reduces losses [253].

The spraying method is used for coating Mozzarella cheese by electrostatic spraying with alginate, chitosan, and soy protein isolate solutions [255], freshly cut lotus roots by pressure atomization with xanthan gum [256], strawberry by electrostatic and usual spraying with sodium alginate [257], and products by electrostatic spraying with chitosan solutions of different deacetylation degrees [258].

### 3.3. Edible Films Composition

The most important properties of edible films are density, thickness, transparency, degree of swelling, thermal stability, mechanical strength, barrier properties to oxygen, and water vapor and safety [259]. The possibility of adding ingredients to the composition of edible films for improving the physico-chemical, mechanical, optical, and microbiological parameters compared with their basic compositions is of great interest to researchers [72,245]. Auxiliary components include plasticizers, crosslinking agents, and other additives [260].

The addition of a plasticizer is an important step in the preparation of solutions from proteins and polysaccharides, which allow the formation of a brittle structure of edible films to be excluded [186]. Most plasticizers contain hydroxyl groups that form hydrogen bonds with biopolymers and increase the free space and flexibility of the polymer matrix [261]. Water, mono-, di-, and oligosaccharides (glucose, sucrose, and fructose) [262], polyatomic alcohols (glycerin and sorbitol) [263], and lipids and their derivatives (lecithin, oleic acid, and waxes) [264] are the main plasticizers.

Crosslinking agents are used to improve the hydration, mechanical, and barrier properties of films and coatings based on proteins and polysaccharides [265]. The process of crosslinking consists of the formation of a three-dimensional structure of the solution by binding the polymer chains with covalent or non-covalent bonds, which leads to an increase in the hydrophobicity of the biopolymer [4]. The choice of crosslinking agents depends on the chemical structure, the presence of active groups, and the molecular weight of the biopolymer, as well as its compatibility with the crosslinking agent [266]. Crosslinking agents and plasticizers must be safe and approved for use in the food industry [267].

#### 3.3.1. Edible Starch Films

Starch is used to produce biodegradable films due to its ability to form a continuous matrix and low permeability to oxygen [268]. The production of edible films from starch is based on the process of gelatinization, i.e., the destruction of the native structure of the biopolymer. Gelatinization is accompanied by a retrogradation process, which includes a reduction of the amount of water-soluble substances in the dissolved starch [269]. The resulting mixture consists of solubilized amylose and partially residual amylopectin granules [270]. Amylose is more prone to retrogradation, the rate of which depends on the amount of starch in the film, the ratio of amylose:amylopectin, their structure, etc. [271]. It leads to undesirable changes in the mechanical and thermal characteristics of the film [272].

A sufficient amount of water (at least 65% of the biopolymer mass) and exposure to high temperatures ranging from 60 to 95 °C are important conditions for hydration of starch molecules for the film processing [273,274]. Water is a good plasticizer, but it is often not enough to obtain the necessary elasticity of the film [275]. The evaporation of water during the drying process can lead to increasing fragility of the starch films [276]. It is necessary to use a plasticizer, such as citric acid and glycerin [277,278], and/or modified starch to reduce its hydrophilic properties [175,279]. The characteristics of starch films depend on the source of the starch, the content and type of plasticizer, and the processing conditions [175]. The high content of amylose in starch results in an elevation of the strength properties of the obtained films since amylose can easily form crystallites and new bonds [269].

Nawab et al. [280] used starch plasticized with glycerin and/or sorbitol from mango seeds for coating of nuts. Mehyar and Han [281] studied mechanical and barrier properties of rice and pea starch films with a high content of amylose and plasticized with glycerin. The properties of cassava starch with glycerin, citric acid, and without it were investigated by Chiumarelli et al. [282] for preserving the quality of mango during the entire shelf life.

#### 3.3.2. Edible Cellulose Films

Cellulose is insoluble in polar solvents, such as water, alcohols, etc. [283], due to its complex and dense linear structure, which includes a large number of intra- and intermolecular hydrogen bonds [284,285], as well as its ability to crystallize [286]. The solubility of cellulose is assumed to be related to its amphiphilic nature and hydrophobic interactions in the polymer structure [287,288].

A lot of studies has focused on finding new ways to dissolve cellulose using the following unusual solvents: N-methylmorpholine N-oxide (NMMO) [285], ionic liquids (a group of organic salts in the liquid state) [289], non-aqueous solvent LiCl/N,N-dimethylacetamide (LiCl/DMAc) [290], aqueous solution of NaOH [291], aqueous solution of alkalis/urea and aqueous solution of NaOH/thiourea [292], mixture of tetra-n-butylammonium fluoride trihydrate (TBAF×3H_2_O) and dimethyl sulfoxide (DMSO) [293], hydrates of inorganic molten salts, as an example ZnCl_2_×4H_2_O [294], and aqueous solutions of metal complexes consisting of transition metal ions and nitrogenous ligands, the most well-known of which are cuprammonium hydroxide (Cuam) and bis(ethylenediamine) copper (II) hydroxide (Cuen) [295]. 

The physical regeneration of a cellulose solution with coagulants (anti-solvents or non-solvents) is one of the methods of obtaining films and coatings [285]. Zhang et al. [296] used H_2_SO_4_, CH_3_COOH, H_2_SO_4_/Na_2_SO_4_, Na_2_SO_4,_ etc., as coagulants for following cellulose regeneration in NaOH/urea. Geng et al. [297] studied a mixture of acetone with water for cellulose precipitation with following regeneration in NaOH/urea/water. The obtained films have good mechanical and barrier properties [298]. Cellophane and cuprofen are the main products prepared from regenerated cellulose and used in the food, medical, and other industries. However, they are losing popularity due to the high cost of production, as well as low barrier properties [299].

Films made of cellulose derivatives dissolve in polar solvents, including water, alcohols, etc., and, therefore, are of interest to researchers [252]. Cellulose derivatives have good mechanical and barrier properties to lipids and gases, including oxygen (O_2_) and carbon dioxide (CO_2_) [300]. The production of films from carboxymethylcellulose (CMC) and cellulose nanocrystals (CNC) is quite common. Cellulose nanocrystals or nanocellulose is a highly crystalline material with improved thermal, mechanical, and barrier properties, and is obtained by acid hydrolysis [301]. It is noted that the nanofiber of CMC has a negative charge and forms a film in the hydrogel state due to ion crosslinking [302]. Thus, Oun and Rhim [303] studied CMC films plasticized with glycerin with or without CNC from rice, barley, and wheat straw. Li et al. [304] used CMC with CNC from pea husks, and Singh et al. [305] used sodium CMC and hydroxyethylcellulose with citric acid as a crosslinking agent. A coating of blueberries from CMC were prepared with the addition of plasticizers—sorbitol and propylene glycol [306].

CNC in a matrix with another biopolymer form a percolated grid linked by hydrogen bonds [307,308]; therefore, cellulose is used as a reinforcing component [309], for example, in films of oxidized starch [310] or pea starch with CMC hydrogel [311,312], mango puree [313], and plasticized gelatin [314]. Moreover, CNC could be used for improving the barrier properties of chitosan films to water vapor [315,316].

#### 3.3.3. Edible Pectin Films

Pectins are soluble in acid and water, most often used as a gelling agent [317]. Many studies consider homogalacturonan, rhamnogalacturonan I, and rhamnogalacturonan II the main monomers of pectin [318]. Pectins have a large number of negatively charged carboxyl groups in the chemical structure and exhibit the ability to interact with metal cations and can bind the active ingredients [319]. 

The molecular weight of the biopolymer, the degree of esterification, and acetyl esterification depend on the source and extraction conditions, and affect the gelation, texture, and stability of the pectin systems [320]. The degree of methoxylation of galacturonic acid residues is the main factor determining the functionality of pectin [321]. Gelation of low-methoxylated pectin (DM < 50) involves electrostatic interactions between cations and negatively charged regions of polymer chains [322]. This model of interaction is called “egg box”—ionic bonds through calcium bridges between carboxyl groups [323]. The stabilization of the system between two adjacent chains of “egg boxes” can occur by van der Waals forces, hydrogen bonds, and electrostatic interactions [321]. Amidated pectin is characterized by a low content of methoxyl groups. It requires less calcium ions to form a gel, is more stable to pH fluctuations compared to non-amidated pectin, and as a result, it is less sensitive to precipitation with a large amount of calcium, and its gels are thermally reversible [72]. Pectin with a high content of methoxyl groups (DM > 50) has a gel-forming ability in an acidic pH < 3.5 in the presence of cosolute with a high concentration, which promotes hydrophobic interactions between methoxyl groups by reducing the activity of water [321]. A low pH is necessary to reduce the dissociation of carboxyl groups, which can form hydrogen bonds with secondary alcohol groups [323].

It is known that pectin gels can be formed due to the formation of covalent cross-links [317], e.g., feruloylated pectins have ferulic acid in the side chains, which can be cross-linked using the oxidative enzymes laccase or peroxidase and form a covalently bound structure, since they have poor gelling properties [324]. The functional properties of such gels differ from those formed by other bonds: the rigidity of the gel increases and the elasticity decreases [321]. 

Pectin films have good mechanical and barrier properties to oxygen, high resistance to fats, but low stability to moisture [325]. Pectin films have a brittle structure, while the addition of a plasticizer does not always lead to high elasticity, so the use of crosslinking agents has a positive effect on the mechanical characteristics of such films [326,327]. Low pectin methoxyl from the cocoa peel was plasticized with glycerin/sorbitol [328], pectin film without additives was used for freshly cut pears [329], calcium lactate as a cross-linking agent was added to low pectin methoxyl for coating of melon [330], while high pectin methoxyl was plasticized with sorbitol for mango dipping [331].

#### 3.3.4. Edible Chitosan Films

Chitosan has good film-forming properties and is characterized by an increased viscosity of the solution during hydration [332]. Antimicrobial activity is one of the features of chitosan due to the presence of active amino groups in the structure [333]. The biopolymer is insoluble in water and usual organic solvents [334]. Its solubility depends on the degree of deacetylation, the location of acetyl groups along the main chain, and the molecular weight. Chitosan is soluble in dilute acid solutions at a pH below 6.0–6.3 due to the presence of amino groups [335]. The mechanical characteristics of the film improve when increasing the molecular weight of the biopolymer [336]. The polycationic properties of chitosan allow the formation of films by breaking the fragments of the polymer chain and then converting them into a film matrix or gel, for example, by evaporation of the solvent, creating hydrophilic and hydrogen bonds and/or electrolytic and ionic crosslinking [337]. Chitosan can effectively improve the mechanical and barrier properties of the film due to the electrostatic attraction [338].

Chitosan-based films are transparent, flexible, and durable [339], have good resistance to fats, selective permeability to carbon dioxide (CO_2_) and oxygen (O_2_), but are very sensitive to moisture [340,341]. Chitosan is also used to enhance the emulsifying effect, reduce the acidity, and stabilize the color of the films [342]. Chitosan is not a thermoplastic material, as it decomposes to the melting point. Therefore, it cannot be extruded, and the films cannot be heat-sealed, but this problem is solved by using it in conjunction with thermoplastic polymers [343].

Chitosan was applied as the main biopolymer for creating edible films in many studies. Chitosan plasticized with glycerin was used for films production [335,344], a film-forming solution of chitosan in a 1% solution of acetic acid was used for dipping sardines [345] and its mixture with glycerin was used for packaging Mozzarella cheese [346] and rainbow trout together with high hydrostatic pressure technology [347], and a film based on chitosan dissolved in deionized water with the addition of glacial acetic acid was wrapped around the fillet of sea bass [348].

#### 3.3.5. Edible Alginate Films

Alginate is a linear polysaccharide with moderate branching. Due to this, it is able to form films with high strength [349]. Sodium, potassium, and ammonium alginates are highly soluble in water, while alginic acid and calcium alginate are not [97]. This anionic polysaccharide with carboxyl groups in each monomer block can react with cations of polyvalent metals (calcium, magnesium, manganese, aluminum, iron, etc.), which makes it possible to obtain strong water-resistant films [350].

The composition, sequence, and ratio of alginate monomers M (β-d-mannuronic acid)/G (α-l-guluronic acid) directly affect the properties of alginate, alginate gels, and films [351]. A large number of G-monomers indicates the ability to form strong films, while the predominance of M-monomers often leads to more elastic films [352]. The unique property of alginates is based on the interaction with metal cations, especially with calcium ions [353]. The ions form bonds between the M and G blocks, resulting in a stable and ordered three-dimensional “egg box” model grid [354]. The formation of bonds occurs due to the length and selectivity of the G blocks, and the M and MG blocks are practically devoid of selectivity [355]. The crosslinking process is used to improve resistance to water, mechanical strength, cohesion and rigidity, and the ability to retain active components in a polymer matrix [244]. The process of interaction with ions is rapid; therefore, a fast-setting and thermally stable gel or film is formed, as a result of which the homogeneity of the resulting material can be disrupted, so it is important to control the rate of ion release [295].

Alginates form transparent, homogeneous, water-soluble films with high resistance to fats and low permeability to oxygen [353,356]. In a number of studies, alginates are used as the main component of biofilms; for example, the ratio of components was selected using the surface response methodology in films made of sodium alginate with glycerin crosslinked with calcium chloride and citric acid [357]; sodium alginate crosslinked with calcium chloride [354,358], and sodium alginate plasticized with glycerin [196]; alginate films plasticized with various types of compounds—PEG-8000, glycerin, fructose, or sorbitol [262].

#### 3.3.6. Edible Casein Films

Caseins can bind ions and small molecules, have outstanding surface-active, stabilizing, and emulsifying properties, as well as good gel-forming and water-binding ability [359]. The preparation of edible films from aqueous solutions of caseinate occurs by its solubilization due to the disordered structure of the protein’s helix [360]. However, the effect of a buffer solution with pH 4.6 leads to the formation of water-insoluble substances, since the protein is in the isoelectric point [361,362].

Caseinates can form intermolecular hydrogen, hydrophobic and electrostatic bonds, which lead to increased interactions between the chains [363]. The structure of the biopolymer matrix is stabilized by hydrophobic, ionic, and hydrogen bonds [115], while binding with hydrophobic molecules occurs by hydrophobic interactions, Van der Waals forces, and hydrogen bonds [362].

Milk proteins form elastic transparent films without odor [113]. The casein films demonstrate resistance to denaturation and/or coagulation and remain stable in a wide range of temperatures, pH, and salt concentrations [364]. Casein-based films for food packaging have good mechanical properties due to the presence of calcium ions and salts of other metals, low oxygen permeability, poor elasticity, high sensitivity to moisture, as well as allergenicity [183,365]. Calcium caseinate films have better barrier properties but are more rigid, whereas sodium caseinate films have better optical properties and elongation under tension [363,366].

The application of caseins in the production of films is quite widespread, for example, films from casein cross-linked by transglutaminase [365] or tannic acid [114], calcium or sodium caseinates and β-casein isolate plasticized with glycerin [115,367,368], films based on casein treated with carbon dioxide [369], or cold plasma with dielectric barrier discharges for improving the functional properties of the films [370].

#### 3.3.7. Edible Collagen Films 

Type I collagen has a fibrous structure and is most often used for films processing [371]. The main amino acid composition of collagen includes glycine, proline, and hydroxyproline [372]. The amino acid composition and molecular weight affect the type and number of interactions involved in the stabilization of the protein matrix, which include disulfide covalent bonds, hydrogen bonds, electrostatic attraction forces, and hydrophobic interactions [373]. A hydrogen bond is formed between the glycine and the amide group in the adjacent chains, which is the main factor in the stabilization of the collagen triple helix [374]. Collagen is a hydrophilic protein, since it contains more acidic, basic, and hydroxylated amino acid residues than lipophilic ones [375].

Collagen has two main properties: gelling, which includes gelation, thickening, etc., and surface activity, including emulsification, adhesion and cohesion, film-forming ability, etc. [145]. The collagen film has good barrier properties to moisture, oxygen (O_2_), prevents the migration of dissolved substances, provides structural integrity and vapor permeability, but has disadvantages, including a rough surface and low thermal stability [376]. The low mechanical strength of the collagen films often leads to breaks during packing [375]. Crosslinking or aging treatments are two ways to solve these problems [377]. There are many additives that can improve the physical and chemical properties of films by chemical crosslinking—formaldehyde, carbodiimide, hexamethylene-1,6-diaminocarboxysulfonate, glutaraldehyde, heparin, transglutaminase, etc. [375], but most of them are cytotoxic [377]. Aging treatments is considered a method of physical crosslinking accompanied with changes in orientation of the collagen fibers, for its implementation, temperature, humidity, and time parameters can be used. The optimal conditions for improving the physical and chemical properties, such as mechanical parameters and reduced water absorption capacity, could be obtained by varying the aging parameters of the collagen films [375].

Collagen-based films have also been used in the food industry for a long time. For film processing, the extracted collagen is pre-treated with an acidic and/or alkaline solution [378], and the collagen fiber plasticized with glycerin [379] or lyophilized collagen are crosslinked with transglutaminase [380]. Many meat products and feed are packed in a collagen film [381].

#### 3.3.8. Edible Gelatin Films

Gelatin is a hydrolyzed form of collagen [176]. The hydrolysis conditions (catalyst and its concentration, the process temperature and duration, etc.) determine the functional characteristics of the hydrolysate [153]. Collagen consisting of f-chains with a molecular weight of about 100 kDa can be dissolved in acid during hydrolysis without changing its original triple helix configuration. Non-covalent bonds are broken during this process [382]. The breaking of both hydrogen and covalent bonds occur during subsequent heat treatment; this leads to the destabilization of the triple helix as a result of the helix-helix transition [383] and the transformation into soluble gelatin [384]. The amino acid composition and molecular weight are the main factors affecting the physical and structural properties of gelatin, which form the mechanical and barrier properties, and the thermal stability of the obtained films and gels [384,385].

Gelatin demonstrates a film-forming ability and forms transparent, slightly colored films with high elongation [386], can form large grids due to its large molecular weight, and incorporate various components [314]. Gelatin-based materials have some serious application limitations due to their high sensitivity to moisture, solubility in water, poor mechanical properties (brittleness), and low thermal stability [387,388]. Hydrophobic plasticizers, such as citric acid esters, can be used for improving the barrier properties to water vapor [389]. Gelatin is dissolved in hot water and the solution is poured into a container and dried to obtain an edible film [390]. The edible films become denser, and the mechanical properties improve with an increase in the protein content, but the permeability to water vapor decreases [27]. 

Plasticized gelatin from farm animal and fish skin [391,392] or gelatin cross-linked with alginate dialdehyde are used for edible film processing [393]. Sorbitol [394] and triacetin [395] also could be used as plasticizers for films production from cattle and pork gelatin.

### 3.4. Ways of Edible Films Improving, Production and Application in Food Pachaging Based on Biopolymer Properties

Animal proteins and natural polysaccharides have both advantages and disadvantages caused by its chemical nature and structure. Taking into account these characteristic, undesirable properties could be improved, while positive ones could be considered for films and coatings production for each biopolymer and type of food product. The summarized information is presented in Table 1.

Edible films based on one biopolymer are often characterized high sensitivity to moisture and and poor barrier to water wapor due to its hydrophilicity nature, as well as brittleness and insufficient mechanical properties. These disadvantages could be solved by different modifications, cross-linking agents and plasticizers addition, and especially by creating composite films and blending.

### 3.5. Edible Composite Films

The application of various combinations of biopolymers is of interest to researchers because it can eliminate the disadvantages of single-component films and achieve the desired barrier and physical-mechanical characteristics of edible films [163]. The properties of edible films and coatings can be changed based on the hydrophobic-hydrophilic characteristics of certain biopolymers [244]. Hydrophobic molecules have a positive effect on the barrier properties to moisture [456]. Oppositely, hydrophilic molecules contribute to the production of materials with higher strength characteristics and low gas permeability [457]. Table 2 shows some variations of edible films obtained by mixing different types of biopolymers and auxiliary ingredients to improve the properties of composite films.

The knowledge of the nature, structure, and properties of biopolymers allows compositions to be combined for producing films and coatings with improved properties [23]. Composite materials are obtained in the form of a multilayer structure by alternating hydrophilic and hydrophobic layers, or in the form of single-layer films with a multicomponent composition [486].

### 3.6. Safety Requirements for Components of Edible Films

Materials in contact with food containing food additives, ingredients, and agents should not have a negative impact on the properties of the product and must meet current hygienic requirements [487]. Nowadays, there are no relevant legislative documents for biopolymer films and coatings, so manufacturers are guided by regulatory documents related to polymer materials in contact with food products [441].

In accordance with current manufacturing practices, edible packaging materials must have a permit from the Food and Drug Administration (FDA), after which they are usually qualified as Generally Recognized as Safe (GRAS) products [488]. GRAS materials can be used in edible films and coatings within the limits set by the FDA [489]. If the material does not qualify for GRAS, the manufacturer may apply for a safety confirmation of the component used or use it without approval, which applies to materials that received "prior approval" or were entered before 1958 in the FDA register [490]. The main risk for the use of components is the possible migration of substances from the packaging to the food [441]. 

Except for chitosan, polysaccharides, including cellulose and its derivatives (CMC, MC, HPMC), starches and its derivatives, pectins, and seaweed extracts (agar, alginates, carrageenan), are either approved food additives or GRAS substances [388]. The most commonly used proteins, such as corn zein, wheat gluten, soy protein and milk proteins, some types of collagen, and gelatin have GRAS status [491]. However, there are some concerns about possible allergenicity or intolerance of wheat or milk proteins for some consumers [492].

Edible food films and coatings are also biodegradable, which implies their complete decomposition by microorganisms during the composting process to products that do not harm the environment [492]. The problem with biodegradable films and coatings is that they must perform their functions safely and effectively for a certain period of time, and only then must they undergo a decomposition stage [490].

## 4. Methods for Estimating Edible Films Quality

Edible films are considered a special type of packaging material that can be eaten together with the product [237]. Such types of biopolymers as proteins, polysaccharides and lipids, as well as their various combinations, are used for edible films production [493]. The composition and structure of biopolymer films, as well as the introduction of various functional components, have a significant impact on the properties of the obtained material. In this regard, there is an obvious need for a comprehensive assessment of the properties of the developed film materials for their further targeted application. The main parameters include mechanical, barrier, optical, hydratative, and microstructural parameters [494]. A comprehensive study of the listed characteristics of the films is possible due to the use of modern research methods discussed in this section.

### 4.1. Basic Requirements for Testing the Condition of Films

Most researchers rely on the requirements of existing regulatory documents developed for synthetic polymer materials when studying the properties of biopolymer films. In this case, it is possible to adapt standardized methods depending on the equipment and the specific test conditions in a certain laboratory. 

Special attention should be paid to maintaining the same conditions before and during the testing of films in order to achieve comparability and reliability of the obtained results. The ASTM D882–10 standard establishes the recommended sample preparation requirements for the following parameters: temperature 23 ± 2 °C and a relative humidity of 50 ± 10% for 40 h [495]. Previously, the films are placed in a desiccator containing a saturated solution of magnesium chloride MgCl_2_ [29], magnesium nitrate (Mg (NO_3_)_2_) [496], or sodium bromide (NaBr) [452], etc. Conditioning is necessary, since most edible materials tend to absorb excess moisture, which can lead to changes in the mechanical, physico-chemical, and hydration properties of the films [497]. 

An important parameter for testing is the precise determination of the thickness of the material, since some properties, such as mechanical or physico-chemical, depend significantly on this indicator. The film thickness is measured with a manual digital or mechanical micrometer with an accuracy of 0.001 mm in several repetitions, choosing arbitrary points in the center and around the perimeter [29]. The average value of the obtained results is taken as the film thickness.

### 4.2. Mechanical Properties

The mechanical strength of the edible film is one of the most important characteristics that allow the limit of the material’s resistance to physical destruction to be determined. The value of the mechanical strength indirectly indicates the degree of food product protection from physical and chemical impacts [498]. 

The determination of mechanical properties includes investigation of such indicators as tensile strength (TS), elongation at break (EB), elastic modulus (EM), and puncture force (PF). The tensile strength, elongation to break, and elastic modulus are the most common parameters used to evaluate mechanical properties [499].

The determination of TS and EB is carried out according to standards ASTM D882–10 and ISO 527–1:2019 [495,500]. Universal breaking machines or texture analyzers are usually used for testing edible films and providing a constant rate of force increase with a relative error of the breaking force of ±1.0 % and an absolute error of the elongation of ±1.0 mm [495].

The equipment must be fitted with tension clamps, which may be metal or in the form of sponges [401]. The use of cardboard gaskets to avoid slipping and cracking of the film in the clamps is allowed [501]. The ends of the gaskets must be at the level of the clamping planes, which limit the clamping length of the sample.

After conditioning, rectangular strips of a certain standard length and width are cut out from each film sample using a scissor [29], scalpel [502], blade [501], or guillotine [503]. According to the standard, the nominal width of the test samples is not less than 5.0 mm and not more than 25.4 mm [495]. The prepared samples must be free of visible damage, tears, and notches at the edges. The film is fixed in a strictly vertical position in the clamps with a set distance. The pre-force value is selected depending on the thickness (structure) of the films. The initial distance between the clamps and the required speed of the crosshead are set before testing. A breaking machine or texture analyzer records values using an appropriate software [504].

Tensile strength (TS, MPa) is calculated by dividing the maximum force on the initial cross section of the sample using the equation [505]:(1)TS = Fmax/A
where Fmax is the maximum force (H) and A is the sample cross-sectional square (m^2^).

Elongation at break (E, %) is calculated using the equation [506]:(2)E =(b/a)×100
where b is the sample length during deformation (mm) and a is the initial sample length (mm).

The calculation of the obtained data is carried out to the first decimal place, followed by rounding to an integer. The measurements are repeated at least 5 times, and the arithmetic mean value is calculated.

Elastic modulus (EM) is the characteristic of film rigidity. The equipment and the method of preparing films for testing are similar to the determination of tensile strength and elongation at break [507]. The elastic modulus is determined by the ratio of the normal force to the corresponding deformation within the limits of proportionality [508]. 

The EM (MPa) could be determined according to the angle of the curve tension-deformation using the equation [509]:(3)$$\;\mathrm{EM}\;={\;\sigma }_{T}/\left(\varepsilon_{T}\times \exp\left(–\varepsilon_{T}\;*k\right)\right)$$
where ε_Τ_ and σ_Τ_ are the deformation and tension, respectively, and k is a constant coefficient.

Puncture force determines the resistance of the material to dynamic puncture and the spread of tear when making a puncture. The measurement of this indicator is carried out according to ASTM D6241–14 standard [510].

Studies of films are carried out on testing machines, for example, texture analyzers, equipped with special nozzles designed to measure the resistance to puncture. The film samples are pre-condensed before testing. Films of a certain shape and size, for example, 30 mm × 30 mm, are placed on the support ring of the texture analyzer. The round aluminum plate is fixed with screws to prevent the films from slipping off. Then, a spherical stainless-steel probe of a certain diameter is brought perpendicular to the surface of the film at a constant speed of 1 mm/s until it punctures the sample. The puncture force coefficient is obtained from the “force-deformation” curves recorded using the software [496]. Puncture deformation (PD, %) is calculated using the equation [511]:(4) PD =(√(D^2–lo^2–lo )/lo)×100
where l_o_ is the initial length of the film, equal to the radius of the annular space (mm) and D is the displacement of the probe at the rupture point (mm).

### 4.3. Physico-Chemical Properties 

The study of physio-chemical characteristics is carried out in order to determine the barrier properties of edible films, which allow the level of permeability of biopolymer films in relation to water vapor and various gases, such as O_2_ and CO_2_, to be assessed. Permeability is one of the most important transport properties of films and coatings, and depends on both biopolymer nature and such properties, as roughness, porosity, arrangements of unidirectionally aligned fibers, area fractal dimension of pore and tortuosity fractal dimension. Factors such as the temperature or presence of electrolytes could also influence on permeability. Some theoretical models were developed to predict the transverse permeability depending on square, staggered, and hexagonal arrangements of fibers. Moreover, permeability can be expressed as a function of the unit cell aspect ratio, porosity, fiber size, net charge density, and viscosity and electrical conductivity of the electrolyte solution [512,513]. Nevertheless, permeability of films is usually evaluated by standard methods, described in ASTM E96 and ASTM D3985 [514,515]. 

#### 4.3.1. Water Vapor Permeability

Water vapor permeability (WVP) is determined by the gravimetric method according to the ASTM E96 [514]. The method is based on the dependence of the change in the mass of the test sample on time. A weighing bottle with a certain depth and diameter with a screw or conventional lid is used for measuring WVP. The film sample is placed on top of the weighing bottle and fixed with a lid with a rubber O-ring [496]. 

There are two main ways of determining WVP. The most common method is the dry method, which is based on the transition of water vapor through the sample from moist air into a space with a desiccant (silica gel) where the humidity is close to zero [29].

According to the second way (wet method), water vapor from a cup of water, where the air humidity is close to 100%, passes through the sample into air having a lower humidity (about 30%). In this case, the samples are placed in ventilated chambers, providing a gradient of relative humidity between the two sides of the film at a level of 30–100% [471]. 

Water vapor permeability (WVP, g·m/m^2^·s·Pa) is calculated using the equation [29]:(5) WVR =(Δm×e)/(A×Δt×Δp)
where Δm/Δt is the mass of lost moisture per unit of time (g·s^−1^), A is the square of the film subject to moisture transfer (m^2^), e is the film thickness (m), and Δp is the pressure change of water vapor between the two sides of the film (Pa).

#### 4.3.2. Gaseous Permeability 

Gas permeability is an important property of biopolymer materials since each type of biopolymer has a different degree of gas permeability [516]. Oxygen and carbon dioxide have a great influence on the quality and safety of products during storage; therefore, the determination of gas permeability of films requires special attention [517]. 

The measurement of the gas permeability of the films is carried out according to the ASTM D3985 standard [515]. The test is performed on an appropriate equipment: a tester [518] or an oxygen permeability analyzer [519]. Films were cut into 4 cm diameter and covered with aluminum foil on both sides in order to avoid contact of the test gas with other areas of the film. An uncovered space in the form of a circle was left for the equipment cell [520]. During the test, the prepared film sample is placed in the cell of the analyzer or tester, where the film is isolated on both sides. Oxygen or carbon dioxide is supplied on one side, and a neutral gas consisting of 98% nitrogen and 2% hydrogen on the other [519].

Gas permeability against oxygen or carbon dioxide of films (OP or CP, cm^3^/m·Pa) is calculated using the equation [519]:(6) OP =(OTR×A)/∆p
where OTR is the gas transmission rate (cm^3^/m^2^), A is the film thickness (m), and ∆p is the partial pressure difference between the film sides (Pa).

### 4.4. Hydration Properties

Different hydrocolloids are obtained from different natural sources of animal and plant origin, so the films based on them have different levels of solubility. Obviously, films prepared from proteins biopolymers (sodium caseinate, whey protein isolate, gelatin) and plant polysaccharides (CMC, sodium alginate, and potato starch) demonstrated different hydration properties [521,522]. The solubility of edible films is a key parameter for assessing their resistance to solvents, such as water, acid, and alkali [523].

#### 4.4.1. Films Solubility

Most biopolymer films based on protein and polysaccharides without the introduction of crosslinking agents and/or tanning agents have a high solubility in water [523]. In many studies, the solubility of films is determined according to the method proposed by Gontard and Guilbert [523]. The film samples are ground into small rectangular pieces and dried at a temperature of 100 °C for 24 h to a constant mass. The samples are then placed in 100 mL of distilled water, and either hydrochloric acid (pH 4.0) or sodium hydroxide (pH 10.0), for 24 h. Next, the films are removed from the solutions and re-dried at 100 °C for 24 h, and the final mass is fixed.

The approach described by Gontard et al. [524] is used as an alternative method. The film samples are cut into square pieces (2 cm × 2 cm), and weighed and dried in a drying cabinet at a temperature of 105 °C for 12 h. Then, the dry films are re-weighed to determine the initial mass. The film samples are placed in 30 mL of solvent at a temperature of 25 °C and kept at room temperature for 24 h. The remaining parts of the undissolved film are removed and dried at 105 °C for 24 h and weighed to determine the final mass.

Films solubility (FS, %) is calculated using the equation [520]:(7)$$\mathrm{FS}\;=\left(W_{i}-W_{\gamma }\right)/W_{i}\times 100$$
where Wᵢ is the initial dry film weight (g) and Wᵧ is the final dry film weight (g).

#### 4.4.2. Swelling Index

The swelling index allows the level of moisture absorption (absorption capacity) of the edible film to be evaluated. This parameter has a high correlation with solubility [525] and the structure density and thickness of edible films [29], as well as with hydrophilicity of biopolymers used for films processing. 

According to the method by Basiak et al. (2017), the film samples are cut into 2 cm × 2 cm pieces and weighed and immersed in distilled water (25 °C) for 2 min. Wet samples are blotted with filter paper to remove excess liquid and weighed. A similar Equation (7) is used to determine the swelling index [29].

#### 4.4.3. Contact Angle Measurements

A contact angle measurement allows the level of film hydrophobicity to be determined and is based on the sessile-drop technique. The sessile-drop contact angle (θ) is formed due to the interfacial tension, and the vertex of the angle lies at the interface of three phases: γ_sv_ (solid-vapor), γ_lv_ (liquid-vapor), and γ_sl_ (solid-liquid) [499]. The schematic of a sessile-drop contact angle system is presented in Figure 16. The measurement of the sessile-drop contact angle is carried out by determining the tangent (angle) of the liquid drop to the solid surface at the base [526].

A standard liquid is used for the test, for example, glycerin. There are restrictions on the use of water as a standard liquid since food films have high hydrophilic properties, which lead to their rapid swelling [527]. A drop of 1.5–2.0 µL of the test liquid is applied to the horizontal surface of the film, which was in contact with the air at the time of drying [528]. The sessile-drop contact angle is measured on one or both sides of the drop using a digital microscope equipped with image analysis software [499]. The image of the edge angle is taken immediately, within 60 s after touching the liquid on the surface of the film, in order to avoid evaporation, dissolution, and swelling processes [528]. 

The obtained results allow the relationship between the surface tension energies of the three phases to be described, according to the Young equation [529]:(8)γlv cosγθY = γsv – γsl
where γ_lv_ is the interfacial boundary tension: liquid-gas, γ_sv_ is the interfacial boundary tension: solid-gas, γ_sl_ is the interfacial boundary tension: solid-liquid, and Y is the Young’s edge angle.

### 4.5. Scanning Electron Microscopy 

Scanning electron microscopy (SEM) is performed to determine the morphological features of the surface and cross-section of edible films. The study of the microstructure of films allows an expanded understanding of the influence of modifiers on the processes of structure formation to be obtained, with the identification of cracks, porosity, roughness, uniformity, and density of the material structure (Figure 17).

There is a direct relationship between the structure and the mechanical, physico-chemical, and hydration properties of the films. SEM is recommended for composite films, for example, alginate/pectin [467], nanoemulsions [496], or when introducing functional additives, such as essential oils [530] or nanoparticles [531].

The microstructure of the films is studied using a scanning electron microscope. The films are pre-frozen in liquid nitrogen and randomly destroyed for the assessment of cross-section morphology. The films are treated with carbon, placed on an aluminum holder, and fixed to the support with double-sided tape. Gold or gold-palladium is applied to the surface of the samples for making electroconductive properties (visualization), and the samples are observed at an accelerating voltage of 5 to 15 kV and an operating distance of 10 mm [29].

## 5. Conclusions and Future Perspectives

Traditional plastics do not degrade and accumulate in large quantities, which cause significant harm to the environment. The development and application of bioactive packaging systems from environmentally friendly biopolymers is a relevant field of research. 

Animal proteins and natural polysaccharides are the most common biopolymers for the production of edible films and coatings. Nevertheless, such biopolymers have a list of limitations for this application mainly due to its structure and properties, such us brittleness and unsufficient mechanical properties, as well as high hidrophilicity cased low water stability, high moisture sensitivity and water vapor permeability. These disadvantages could be overcome by blending and composite films production, implementation of crosslinking agents and plasticizers, as well as the application of appropriate ways of film production based on the biopolymer structure and type of food product. Novel edible films must meet the requirements of quality and safety.

We made an effort to summarize information about such popular biopolymers from renewable resources as starch, cellulose, pectin, chitosan, alginate, casein, collagen and gelatin, discuss the main advantages and disadvantages, approaches for properties improving, recommended film-forming solution, appropriate form and ways of production of films and coatings, as well as type of packaged food based on the compatibility and nature of a certain biopolymer. Moreover, approaches for films and coating production were also reviewed as in general so and for certain type of biopolymers. We also collected usual methods for assessment the properties of designed films. This information allows the reader to make the first insight into environmentally friendly packaging from nature of polimer till properties of packaging material.

Nevertheless, films based on one biopolymer could be not so competitive to traditional plastics. In this regard, blending and composite materials are particularly relevant, because combination of biopolymers and its modificated derivates could allow create packaging material with properties of plastics without microplastic addition. On the other hand, created new composites should be more deeply studied, because of containing two or more biopolymer chaines. In this regard, additional investigation approaches, such as scanning electron microscopy, could clear the interoperability of biopolymers. Scanning electron microscopy is also essential for investigation of films with introduced functional additives, which are widely used for improving the biological properties of packaging matherials.

Global ecologization dictates its own conditions to the manufacturer; consumer acceptance is also driven in biodegradability and eco-friendliness. Nowadays, due to the efforts of scientists worldwide, the creation of competitively packaging material based on biopolymers is a reality, but not a myth.

## Figures and Tables

**Figure 1 polymers-13-01592-f001:**
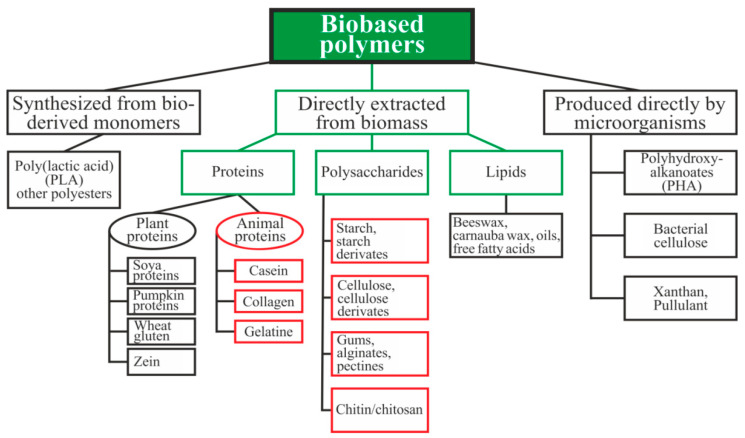
Schematic classification of biopolymer types [15,18,19]. Reproduced with permission from Popović, S.Z. et al., Biopolymers for Food Design; published by Elsevier Inc., 2018.

**Figure 2 polymers-13-01592-f002:**
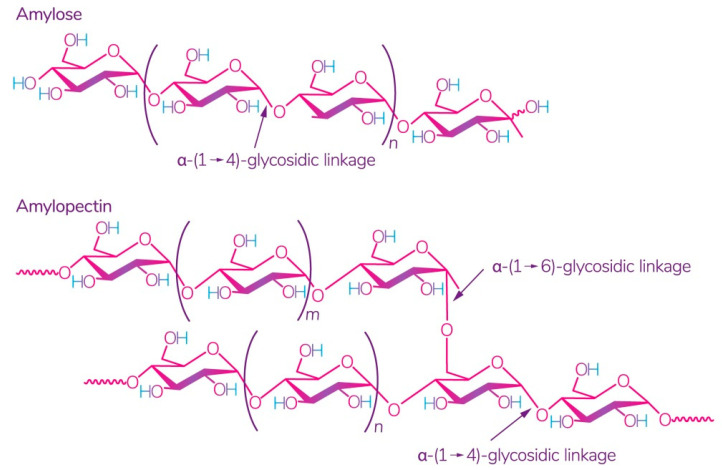
Structure of amylose and amylopectin [30]. Reproduced from Kadokawa, Polymers; published by MDPI, Basel, Switzerland, 2012.

**Figure 3 polymers-13-01592-f003:**
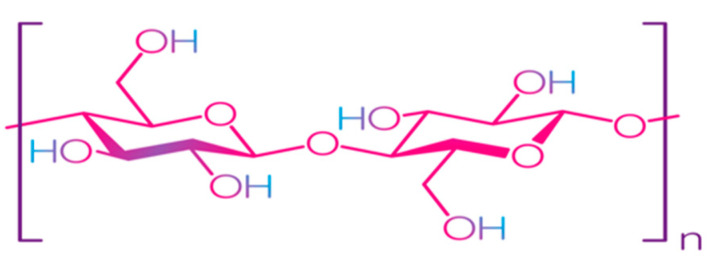
Structure of cellulose [49]. Reproduced with permission from Trache, D. et al., International Journal of Biological Macromolecules; published by Elsevier Ltd., 2016.

**Figure 4 polymers-13-01592-f004:**
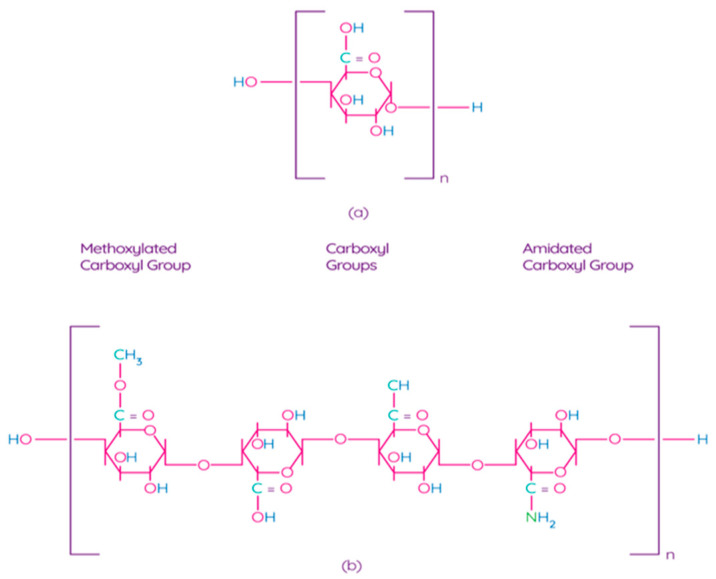
Chemical structure of polygalacturonic acid (**a**) and representative chemical structure of pectin, showing typical repeating groups (**b**) [72]. Reproduced with permission from Espitia, P.J.P. et al., Food Hydrocolloids; published by Elsevier Ltd., 2013.

**Figure 5 polymers-13-01592-f005:**
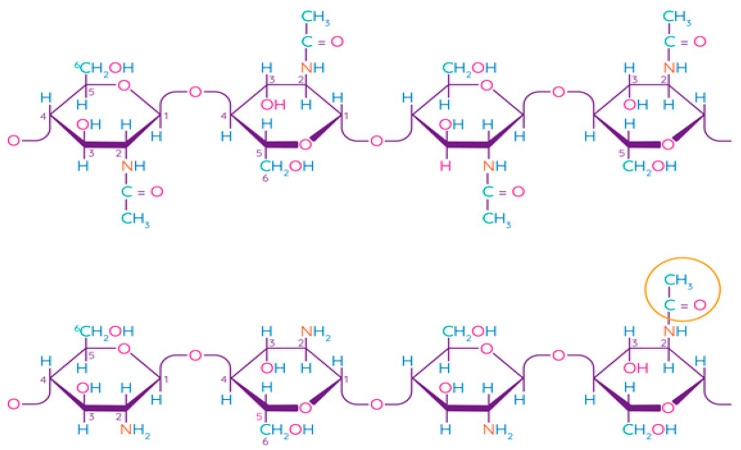
Chemical structure of chitin and chitosan [87]. Reproduced with permission from Rasul, R.M. et al., Carbohydrate Polymers; published by Elsevier Ltd., 2020.

**Figure 6 polymers-13-01592-f006:**
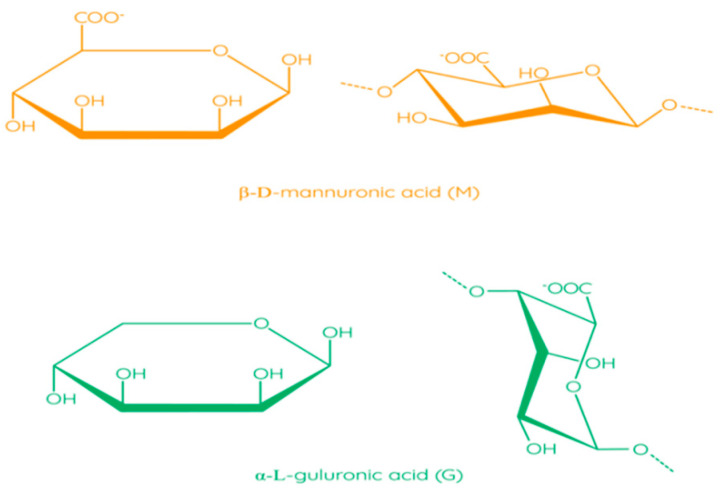
Structural formulas of monomers in alginate [97]. Reproduced from Parreidt, T.S. et al., Foods; published by MDPI, Basel, Switzerland, 2018.

**Figure 7 polymers-13-01592-f007:**
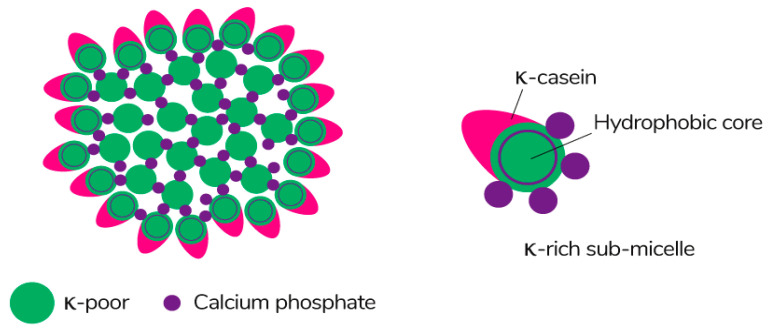
The schematic of the submicelle model of the casein micelle [117]. Reproduced with permission from Horne, D.S., Current Opinion in Colloid & Interface Science; published by Elsevier Ltd., 2005.

**Figure 8 polymers-13-01592-f008:**
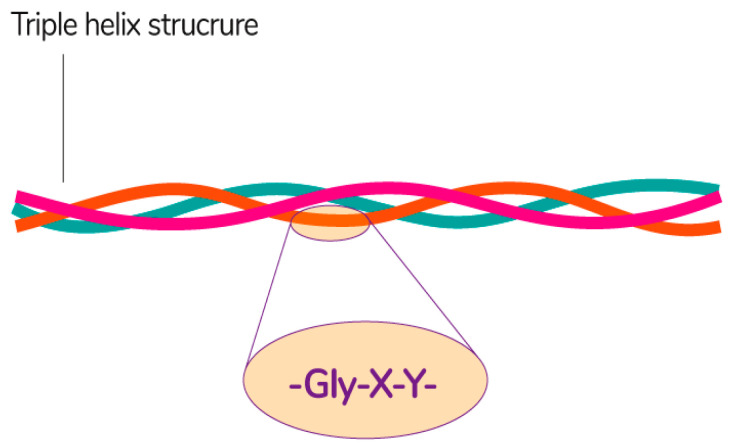
The general structural features of collagen [135]. Reproduced from Sebald, A., proof-read/edited by Mitchell, D.A. Maxfacts; published by University of York, 2019.

**Figure 9 polymers-13-01592-f009:**
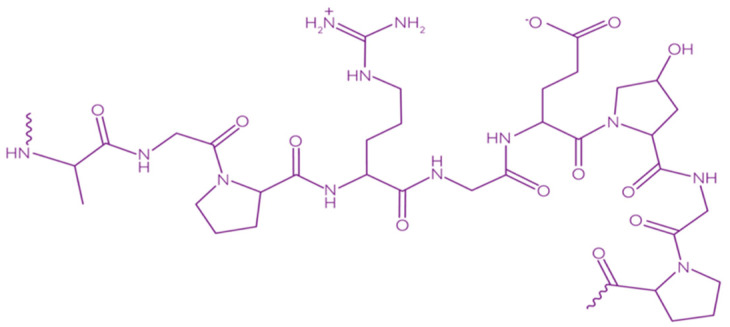
Chemical structure of gelatin [148]. Reproduced with permission from Kariduraganavar, M.Y., et al., Natural and Synthetic Biomedical Polymers; published by Elsevier Inc., 2014.

**Figure 10 polymers-13-01592-f010:**
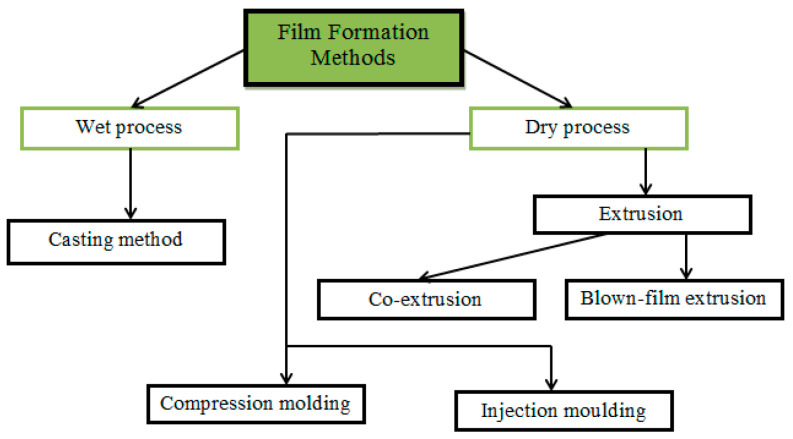
Film formation methods of biopolymers.

**Figure 11 polymers-13-01592-f011:**
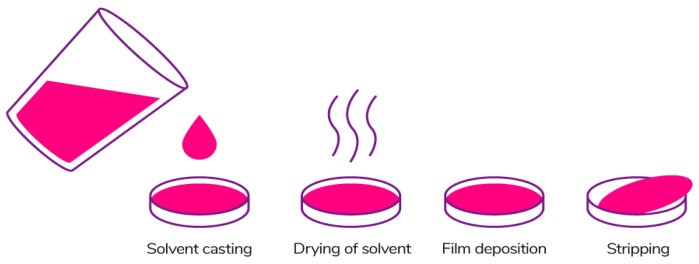
Casting method [186]. Reproduced with permission from Suhag, R., et al., Food Research International; published by Elsevier Ltd., 2020.

**Figure 12 polymers-13-01592-f012:**
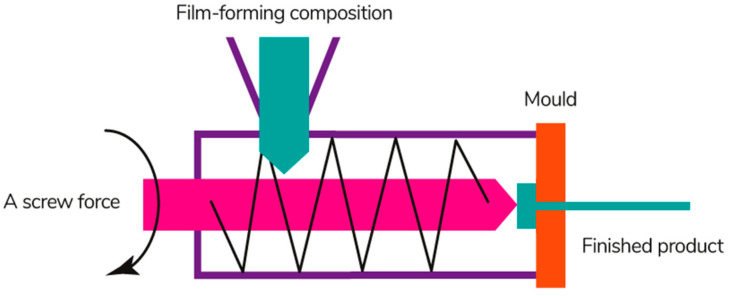
Extrusion process [186]. Reproduced with permission from Suhag, R., et al., Food Research International; published by Elsevier Ltd., 2020.

**Figure 13 polymers-13-01592-f013:**
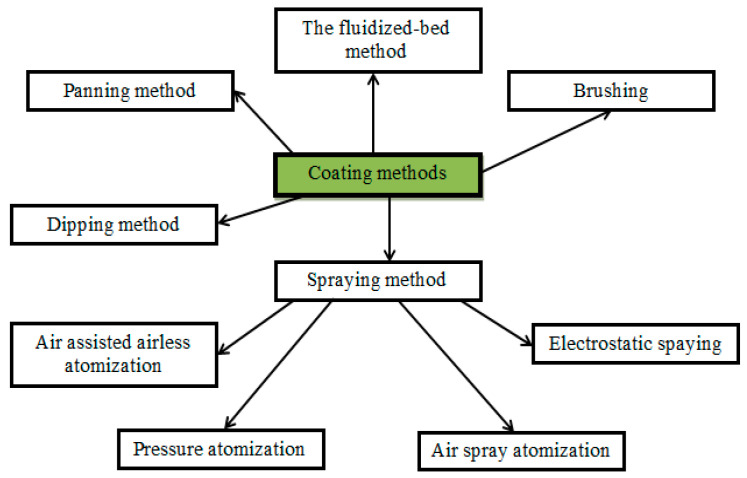
Coating methods.

**Figure 14 polymers-13-01592-f014:**
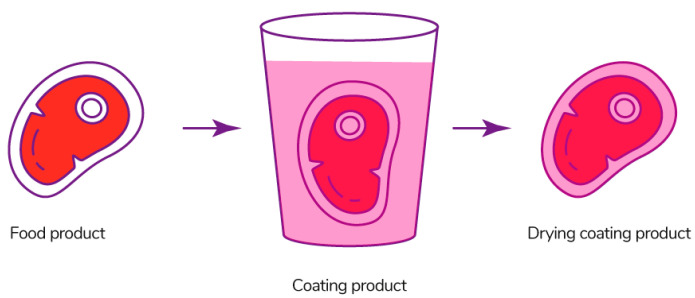
Dipping method [27]. Reproduced with permission from Mohamed, S.A.A., et al., Carbohydrate Polymer; published by Elsevier Ltd., 2020.

**Figure 15 polymers-13-01592-f015:**
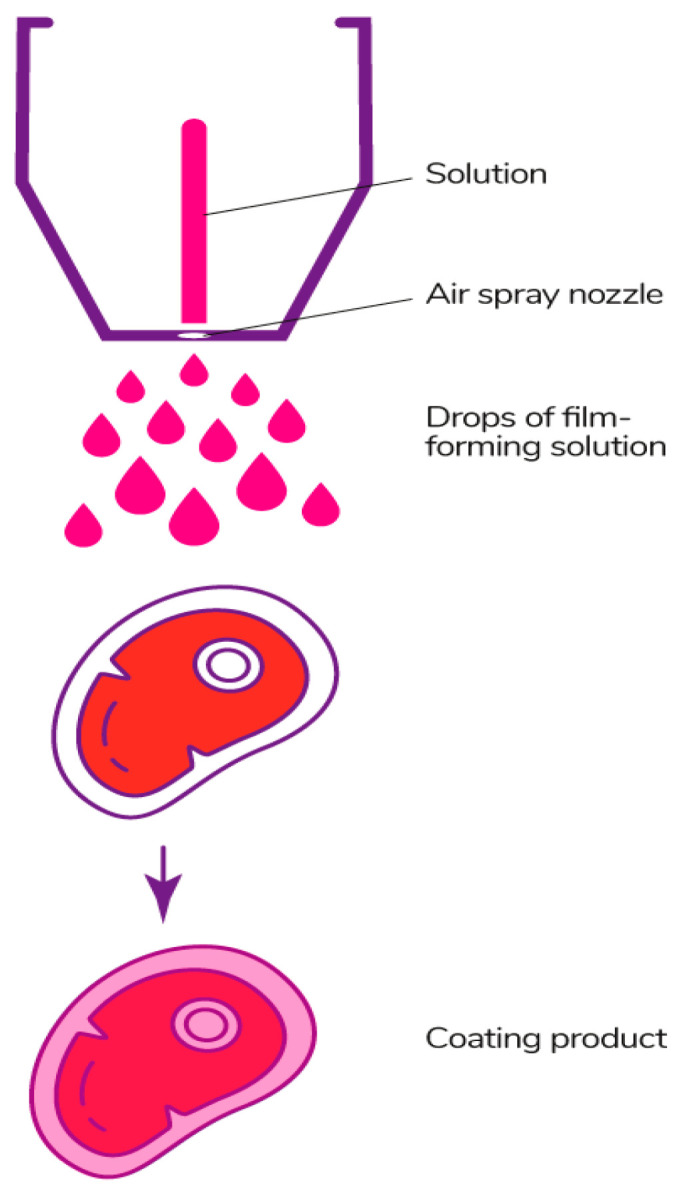
Spraying method.

**Figure 16 polymers-13-01592-f016:**
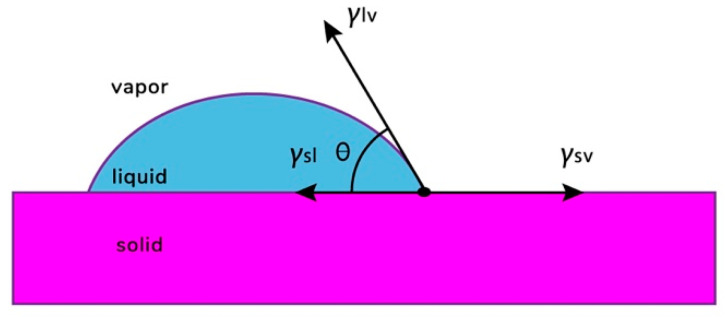
Schematic of a sessile-drop contact angle system [526]. Reproduced with permission from Kwok, D.Y. and Neumann, A.W., Advances in Colloid and Interface Science; Elsevier Science B.V., 1999.

**Figure 17 polymers-13-01592-f017:**
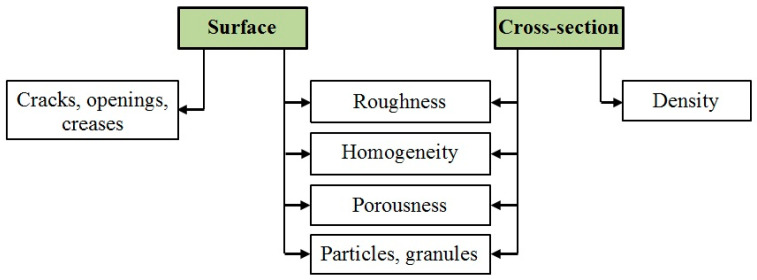
Characteristics of the film structure determined by SEM.

**Table 1 polymers-13-01592-t001:** The main characteristics of packaging materials based on animal proteins and natural polysaccharides and ways of its improving, production and application.

Advantages	Disadvantages	Approaches for Properties Improving	Recommended Film-forming Solution	Appropriate form and Ways of Production	Type of Packaged Food	References
Starch						
Strong and flexible structure Transparency Resistance to fats and oils Good strain at break Tasteless Odorless Very low oxygen permeability	Hydrophilicity Low water stability High moisture sensitivity High water vapor permeability Retrogradation phenomena Brittle behavior at room temperature Poor mechanical properties and processability	Plasticizers addition Compatibilizers addition Structure modification (TPS) Chemical modification (acetylation, hydroxypropylation, acid modified, etc.) Crosslinking (glutaraldehyde, sodium trimetaphosphate, citric acid, etc.) Physical modification (deep freezing and thawing, B-type X-ray diffraction, extrusion heating and fluidized bed heating, ultrasound waves, microwave radiation, osmotic pressure, pulsed electric field, etc.) Gelatinization transition temperatures Enzymatic hydrolysis or modification Elevation of amylose content Micro- and nanosized fillers (starch nanocrystals or nanoparticles) Composite materials and blending	Starch 2–4 wt %, 15–30% of glycerol or sorbitol (mainly glycerol)	Films Wet method (casting) Dry methods (extrusion, injection moulding, and compression moulding) Coatings Dipping and spraying	Fruit, vegetables, berries, meat and some meat products	[175,269,396,397,398,399,400,401,402,403,404,405,406]
Cellulose						
High strength High stiffness Excellent tensile strength and modulus High durability Low bulk Good moldability Resistant to oils and fats Transparent Flavorless Colorless Tasteless High barrier to gases	Infusibility and insolubility Lack of good interfacial adhesion Low melting point Poor resistance towards moisture Poor water vapor barriers Brittleness Sensitive towards pH, temperature, ionic, electro responses	Surface chemical modifications (silylation, mercerization, peroxide, benzoylation, etc.) Cellulose derivates processing (cellulose xanthate, MC, HPMC, HPC, and CMC) Crosslinking (citric acid) Hydrophobization (oxidation, esterification, amidation, carbamination, grafting) Plasticizers and fats/oils/wax addition (glycerol, polyethylene glycol400, palmitic acid, xylitol, sorbitol, etc.) Composite materials and blending	Mainly used as reinforcing agent Composition of film-forming solution strongly depends on type of cellulose derivates, as well as composition of cellulose-containing matherial	Films Wet method (casting) Dry methods (extrusion) Coatings Dipping and spraying	Fruits, berries and vegetables, meat and poultry products with limitations, appropriate for frying oil, breading or batter during frying	[406,407,408,409,410,411,412,413,414,415,416,417]
Pectins						
Good oxygen, aroma, and lipid barriers Transparent Flavorless Colorless Tasteless	Hydrophilicity Subjected to re-dissolution in water or destruction in high humidity conditions Brittle and stiff structure Poor moisture barriers Not strong Poor mechanical characteristics	Plasticizers addition (glycerol, acetylated monoglycerides, poly-ethylene glycol, and sucrose) Crosslinking (ionomers formation, calcium ions) Composite materials and blending	1–3 wt % pectin, 45–50% of glycerol, 1–2% of calcium chloride	Coatings (mainly) Dipping, brushing and spraying Films Wet method (casting)	Fruits, berries and vegetables, meat and poultry products with limitations	[72,78,325,328,406,418]
Chitosan						
Low oxygen and CO_2_ permeability Antimicrobial properties Antioxidant activity Forms strong films Excellent film-forming behavior Compatibility with other substances Good mechanical properties Transparent Flavorless Colorless Tasteless Acts as a chelating agent	High water sensitivity Highly permeable to water vapor Low mechanical and thermal stability Brittleness Stiffness	Addition of neutral lipids, fatty acids waxes and clay Addition of cross-linking agents Chemical modifications (N-methylation, alkylation) Irradiation Ultrasonic treatments Grafting Enzyme treatment Plasticizers addition (glycerol, xylitol, and sorbitol) Thermomechanical treatment Complexation Surface coating Natural deep eutectic solvents s based on choline chloride prepared with malic acid (MA), lactic acid (LA), citric acid, and glycerol Composite materials and blending	1–2% of chitosan (>90% DDA) in 1% acetic acid/malic/lactic/citric acid (mainly acetic acid), 20–30% of glycerol	Films Wet method (casting) Dry methods (extrusion, compression moulding) Coatings Dipping and spraying	Fruits, berries and vegetables, Meat, fish and poultry	[26,46,335,340,406,419,420,421,422,423,424,425,426]
Alginate						
Structural stability Prevent lipid oxidation and stop rancidity Polarity and water-solubility Good heat stability Decent strong film Transparent Flavorless Colorless Tasteless Uniform Good oxygen barriers Good oil barrier properties Good biocompatibility Osteoconductivity	Hydrophilicity Subjected to re-dissolution in water or destruction in high humidity conditions Rigidity Poor acid stability Highly sensitive to calcium ions Tend to form lumps Brittleness Film is easily damaged when dried Weak mechanical strength	Crosslinking (ionic, covalent, photo, enzymatic ) Chemical modification (propylene glycol alginate (PGA), oxidation, sulfation, copolymerization, esterification, amidation, etc) Plasticizers addition (glycerol, sodium lactate, sorbitol, polyethylene glycol (PEG), etc.) Addition of surface active agents (0–5% tween 40, tween 80, span 80, span 60, and soy lecithin) Addition of oils Composite materials and blending	2.5–4% of alginate, 1–1.5% of calcium chloride, 10–50% of glycerol	Films Wet method (casting, more often used) Dry methods (extrusion) Coatings Dipping and spraying	Fruits, berries and vegetables Bakery coatings-icings Meat and poultry, precooked meat products	[26,97,244,354,406,427,428,429,430,431,432,433]
Casein						
Low oxygen and carbon dioxide permeability Mechanical resistance Good strength Adhere well to hydrophilic surfaces High thermal stability Good film forming ability Transparent Desirable flavor Flexible	Brittleness High sensitivity to moisture Hazy films Significant protein particle size Poor barriers to moisture Potential allergenicity Not enough mechanical properties and elasticity	Cross-linking (formaldehyde, glutaraldehyde, lactic acid, genipin, tannic acid, wax) Denaturation Enzyme treatment (transglutaminase) Plasticizers addition (water, glycerol, propylene glycol, and sorbitol) Maillard reaction Changing the ionic strength of the film-forming solution (precipitation with salts, CO_2_CN) pH alteration Physical treatment (photo-induced polymerization, pulsed light, irradiation) Chemical treatment (alkali-treatment) Composite materials and blending	1–10 wt % of casein, 30–50% of glycerol or sorbitol	Films Wet method (casting, more often used) Dry methods (extrusion rarely) Coatings Dipping and spraying (more often than dipping)	Fruits, berries, vegetables, and chees, fish, meat and meat product	[113,115,208,362,406,434,435,436,437,438,439,440,441]
Collagen						
Mechanical resistance Good barriers to oxygen and carbon dioxide Adhere well to hydrophilic surfaces Gel strength Good melting temperature Shape and stability Good mechanical properties	High sensitivity to moisture Poor barriers to moisture Not so strong Prone to rupture Seepage phenomenon Anisotropy Dependence of properties on alignment direction of collagen fiber Poor quality, particularly in terms of strength and elasticity	Cross-linking (gluteraldehyde, carbodimide, transglutarninase, keratin, metal ions) Plasticizers addition (glycerol or sorbitol) UV irradiation High pressure treatment Organic acid treatment Enzymatic treatments (proteases such as papain, bromolain, ficin, fungal protease, trypsin, chymotrypsin, or pepsin) Aging treatment Regeneration Composite materials and blending	3–8 wt % (0–10%) of collagen	Casings or films Extrusion (wet and dry spinning technology) and co-extrusion	Meat and meat product (especially casings for sausages)	[224,406,442,443,444,445,446]
Gelatin						
Good film-forming properties Biocompatibility Transparent Low oxygen permeability Absence of an appreciable odour Transparent Tasteless Gel strength	Low strength Inelasticity Brittleness Limited thermal stability Limited mechanical properties High sensitivity to moisture Poor barriers to moisture	Cross-linking (genipin, transglutaminase, natural extracts, glutaraldehyde) Plasticizers addition (propylene glycol, ethylene glycol, glycerol or sorbitol, Sucrose, oleic acid, citric acid, tartaric acid, malic acid, PEG of different molecular weights, mannitol, EG, DEG, TEG, EA, di ethanol amine (DEA) and TEA) Inorganic and organic materials addition Physical treatment (heating or irradiation) Maillard reaction Chemical modification (acetylation, deamidation, glycation, etc.) Composite materials and blending	2–5 wt % of gelatin, 10–30% of glycerol	Films Wet method (casting) Dry methods (extrusion, blown-extrusion) Coatings Dipping and spraying (dipping is more spread)	Various meat products, poultry, fish, vegetables and fruits	[113,139,406,447,448,449,450,451,452,453,454,455]

**Table 2 polymers-13-01592-t002:** Examples of composites edible films.

Biopolymers	Plasticizer	Crosslinking Agent	Changes in Properties	Reference
Starch				
Cassava and rice starch/maltodextrin/agar	Glycerin	-	High film forming ability for package molding, improved the mechanical and water barrier properties, decreased relaxation temperatures, improved water sensitivity.	[242]
Potato starch/cellulose fibers from sunflower husk	Glycerin	Citric acid	Improved resistance towards stress and sufficient extensibility and high tensile strength, brittleness due to starch-cellulose interactions and decreased starch chain mobility, reinforced network and decreased in swelling.	[279]
Tapioca starch/beeswax/propolis	Glycerin	-	Lower values of water vapor permeability and water solubility; decreased in the moisture content and vapor water permeability.	[458]
Rice starch/cellulose fiber mesocarp	-	-	Enhanced thermal stability and lowered water uptake	[459]
Rice starch/cellulose	Glycerin/sorbitol	-	Reinforced mechanically the films (higher tensile strength) and reduced water vapor permeabilities	[460]
Pea starch/CMC and pea starch/MC	Glycerin	-	Improved the storage modulus and the glass transition temperature, increased the tensile stress, elongation at break and the barrier of water vapor; MC increased the thermalstability, while CMC decreased the thermal stability.	[461]
Turmeric starch/gelatin	Glycerin		Gelatin increased flexibility and elongation at break	[462]
Cellulose and derivatives				
Wood cellulose/sodium alginate	-	Calcium chloride	Increased the mechanical properties (tensile), improved grease barrier properties and reduced water vapor permeability	[463]
Cellulose/collagen hydrolysate	-	-	Exhibited good transparence and the capacity for ultraviolet radiation absorption, improved the mechanical properties and enhanced the stability in distilled water.	[464]
Cellulose/chitosan	-	-	High transparent property, excellent barrier properties against oxygen and antimicrobial properties.	[465]
Pectin				
Fruit and vegetable wastes (fruit and vegetable flour)	-	-	Decreased solubility (50%) and improved of the mechanical properties (decrease of elongation and increase of tensile strength)	[466]
Citrus pectin/sodium alginate	Polyglycerin	Zinc chloride	Improved the strength of crosslinking network, improved mechanical performance.	[467]
Papaya puree/alginate	Glycerin	Calcium chloride/citric acid	Improved puncture strength	[357]
Pectin/protein phaseolin	-	Microbial transglutaminase	Mechanical properties and barrier properties to CO_2_, O_2_ and water vapor was comparable to commercial plastics.	[468]
Chitosan				
Chitosan/collagen	Glycerin	-	Displayed higher elongation at break point, but lower tensile strength and modulus of elasticity, increased water vapor permeability, decreased transparency	[469]
Quaternized chitosan/CMC	-	-	Improved tensile properties, thermostability, oxygen permeability values, and water resistance	[470]
Alginate				
Alginate/pectin	Glycerin	Calcium chloride	Continuous, homogenous and transparent films, chemical composition influenced on color, water vapor permeability, tensile strength, elongation at break	[471]
Alginate/gum	-	Calcium chloride	Improved the strength of network	[472]
Alginate/cotton hydrolysate	Glycerin	-	Increased the barrier properties to visible light, did not affect the moisture content, biodegradability, solubility or oil barrier properties, increased the thickness and water vapor permeability	[473]
Alginate/chitosan	Glycerin	Calcium chloride	Decreased water solubility, but increased film thickness, water vapor permeability and oxygen permeability, good barrier properties against ultraviolet light.	[474]
Casein				
Lactic acid casein powder/carnauba or candelilla waxes	Sorbitol	-	Decreased water permeability	[475]
Casein/cellulose microgel	-	-	Reduced the moisture absorption and the water vapor permeability, homogeneous and dense cross-sectional structure, increased the cleavage temperature, tensile strength and Young’s modulus	[476]
Sodium caseinate/low-methoxylated pectin	-	-	Increased the stiffness of films (Young’s modulus) and decreased flexibility, decreased water content	[477]
Collagen				
Fish skin collagen/chitosan	-	-	Lowered water solubility and lightness	[478]
Cattle skin collagen/HPMC	PEG 1500	-	Elevated thermal decomposition temperature and denaturation temperature, exhibited a more homogeneous and compact structure, improved tensile strength, ultimate elongation, hydrophilicity, stretch-ability and smoothness	[479]
Collagen/galactomannan	Glycerin	-	Convenient values of wettability	[480]
Gelatin				
Gelatin/chitosan	Glycerin	-	No significant difference in tensile strength, thickness and transparency	[481,482]
Soy protein isolate/bovine bone gelatin	Glycerin	-	Increased tensile strength, elongation to break, elastic modulus and swelling property, more transparent, and easier to handle	[483]
Whey protein isolate/gelatin/sodium alginate	Glycerin	-	Improved barrier to oxygen, water vapor and mechanical properties	[484]
Fish gelatin/CMC	Glycerin/sorbitol	-	Increased tensile strength and Young’s modulus, decreased the elongation percent and equilibrium moisture	[485]

## Data Availability

Data sharing not applicable.

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
