# Peer review of "Approaches in Animal Proteins and Natural Polysaccharides Application for Food Packaging: Edible Film Production and Quality Estimation"

_polymers, 2021, doi:10.3390/polym13101592_

Round 1
Reviewer 1 Report
Reviewer #1: polymers-1210276 „ Approaches in animal proteins and natural polysaccharides application for food packaging: edible film production and quality estimation” Andrey Lisitsyn, Anastasiya Semenova, Viktoriya Nasonova, Ekaterina Polishchuk, Natalia Revutskaya, Ivan Kozyrev, Elena Kotenkova *
The review by Kotenkova and colleagues describes the latest advances in the use of natural polymers in food packaging. Most of the literature cited is from the last 20 years. The 472 items cited are almost 20 pages of the manuscript. The literature was well chosen, it includes interesting literature items. The review seems to be generally accurate in interpreting the science from the cited articles, especially in:
- Biopolymers used for food packaging ( 8 biopolymers)
- Approaches for the production of biopolymer-based films and coatings
- Methods for estimating edible films quality ( 5 methods)
The reviewed manuscript is generally well structured. The Polymers' readers are interested in this form of literature review, the choice of journal is correct.
However, authors before resubmission should consider to:
- The information in the polymerization summary is misleading. The authors consider only the sources of natural polymers and their possible modifications. The details of the chemical processes involved are not specified. Therefore, "polymerization" should be omitted in the abstract or all chemical reactions should be accurately described in the text.
- Information in the text should be carefully checked, especially if the authors provide details such as:
Line 230: The sentence “Chitosan (…) and is deacetylated by an alkali derivative of chitin (Figure 5), (…)” should be changed, because alkali is used in the chitin deacetylation reaction. This is the main process of chitin production.
- Line 252: The sentence: “When the acetylation degree (expressed in molar percentages) is below 50 mol.%, the product is called chitosan and becomes soluble in acidic aqueous solutions [87].” should be changed. The acetylation degree (%) in commercial chitosan ranges from 60 to 100%. Below these values, it still dissolves poorly.
- No information about the source of the drawings: Figure 2,3,10,13,15 and 17 and Table 1.
- Line 1055- 1058 – “Authors should discuss the results and how they can be interpreted from the perspective of previous studies and of the working hypotheses. The findings and their implications should be discussed in the broadest context possible. Future research directions may also be highlighted.” This is Author Guide text, should be deleted.
Author Response
Dear Reviewer 1,
Thank You for the reviewing the article.
Please see the attachment.
Best regards,
Authors.

Reviewer 2 Report
Inferior packaging or its absence cause significant food loss (about 20 - 25 %) due to microbiological contamination and oxidative processes, which lead to a decrease in the quality of food products and makes them unsuitable for consumption. The development and application of bioactive packaging systems is a relevant field of research. Application of such smart packaging systems is a tool for protecting food from spoilage and reducing the risk of growth of pathogenic microorganisms due to both the creation of a barrier and the action of active components at the border of the product with the packaging. Currently used packaging materials are mainly produced from petrochemical products. The global problem of environmental pollution makes alternative environmentally friendly and biodegradable polymers to be in demand. Natural biopolymers are an interesting resource for edible films production, as they are environmentally friendly packaging materials. In this paper, the possibilities of the main animal proteins and natural polysaccharides application are considered in the review, including the sources, structure, polymerization, and limitations of usage. The main ways for overcoming the limitations caused by the physico-chemical properties of biopolymers are also discussed, including composites approaches, plasticizers, and the addition of crosslinking agents. Although the topic in this work was interesting, the presentation in this manuscript was very poor. This manuscript should be rejected for published in Polymers. However, if the authors are willing to make the substantial revisions according to my comments, I would be glad to re-review this manuscript. Here are my detailed comments:
- The detailed literature review indicates efforts made by the authors. The coherence of the related work, however, is still not clear. It may help the authors by answering the following questions: Why are these works relevant? Which specific problems were addressed? How are the previous results related with the latest work? What are the outstanding, unresolved, research issues? Which of them has been solved by the proposed study? Answering the questions leads to the novelty of the proposed work naturally. Besides, the current one is nothing but a literature review. Why their work is important comparing to previous reports? I think this is essential to keep the interest of the reader.
- Animal proteins and plant polysaccharides are the most common biopolymers for the production of edible films. Nevertheless, such biopolymers have a list of limitations for this application mainly due to its structure and properties. These disadvantages could be overcome by composite films production, implementation of crosslinking agents and plasticizers, as well as application of appropriate ways of film production based on the biopolymer structure and type of food product. The authors should give some explanation on above conclusions. What the authors meant is not clear to reader.
- Permeability is an important transport property, which is connected with the geometric structure of porous materials. And it is the most significant physical parameters which can influence the flow of gas porous materials. Thus, the permeability is the estimation of the easiness of the flow of gas in porous materials, (see [Fractals, 2019, 27(7): 1950116; International Journal of Hydrogen Energy, 2018, 43(37):17880-17888]). Authors should introduce some related knowledge to readers. I think this is essential to keep the interest of the reader.
- Please, expand the conclusions in relation to the specific goals and the future work.
- Although the results look “making sense”, the authors should dig deeper in the results by presenting some in-depth discussion, such as implications of the results, such as possible application of them.
- English grammar and syntax has to be checked carefully throughout the manuscript. There are several grammatical mistakes in the manuscript and it is very difficult to follow anything if they are not corrected.
Author Response
Dear Reviewer 2,
Thank You for the reviewing the article.
Please see the attachment.
Best regards,
Authors.

Round 2
Reviewer 2 Report
Very good. It is ok.
Author Response
Dear Reviewer 2,
Thank You for the reviewing the article.
Best regards,
Authors.